

# The effect of noise on the stability of convection in a conceptual model of the North Atlantic subpolar gyre

Koen J. van der Heijden[1,a], Swinda K.J. Falkena[1], and Anna S. von der Heydt[1,2]

[1]Department of Physics, Institute for Marine and Atmospheric research Utrecht, Utrecht University, Utrecht, the Netherlands
[2]Centre for Complex Systems Studies, Utrecht University, Utrecht, the Netherlands
[a]Now at: Geophysical Institute, University of Bergen and Bjerknes Institute for Climate Research, Bergen, Norway

**Correspondence:** Koen J. van der Heijden (koen.heijden@uib.no)

**Abstract.** The North Atlantic subpolar gyre (SPG) plays a fundamental role in the Atlantic ocean circulation by providing an important connection between the subtropical Atlantic and the Arctic. It is driven by both wind and density differences that are, in part, caused by convection in the Labrador Sea. Through this deep convection site, the SPG is also linked to the AMOC. There is considerable evidence that this area of convection may diminish or shift in a changing climate. For this reason, the convection in the SPG is considered a tipping point. Here, we extensively study a conceptual model of the SPG to characterize the stability of convection in the gyre. The bifurcation structure of this model is analyzed in order to identify bistable parameter regions. For a range of gyre salinity and freshwater forcing levels the gyre is found to have both convective and non-convective states. Furthermore, noise-induced transitions between convective and non-convective states are possible for a wide range of parameter values. Convection in the SPG becomes increasingly unstable as the gyre salinity decreases and the freshwater forcing increases. However, convection never fully stops and can always restart after a period of no convection. This indicates that, at least in this conceptual model, a collapse of convection in the SPG does not have to be permanent.

## 1 Introduction

The North Atlantic subpolar gyre is a cyclonic ocean circulation that includes parts of the North Atlantic Current, the Irminger current and Labrador current (Li and Born, 2019). As such, it provides a connection between the subtropical Atlantic and the Arctic. The horizontal circulation in the gyre dominates heat and salt transport in the subpolar latitudes of the Atlantic (Born and Stocker, 2014; Klockmann et al., 2020). It is also connected to deep convection sites in the Labrador, Irminger and Iceland basins (e.g. Lozier et al., 2019). Due to these characteristics and its position, the area has been described as 'a key coupling region where vigorous wind systems encounter the southernmost extensions of sea ice and the most variable currents of the North-Atlantic with connections to the deep ocean via convection' (Li and Born, 2019).

The SPG is also connected to the Atlantic Meriodional Overturning Circulation (AMOC, e.g. Spooner et al. (2020)) through its link to deep water formation sites in the Labrador Sea, Nordic Seas, and Irminger and Iceland basins. Conceptual modelling studies have indicated that the collapse of convection in the Labrador Sea could precede a collapse of the AMOC (Neff et al., 2023). The exact nature of the coupling between the AMOC and the SPG is up to some debate, but there is agreement on the fact that the SPG and AMOC are coupled and that their interactions can lead to abrupt climate transitions such as Dansgaard-



Oescher events, during which temperatures at high latitudes in the Atlantic suddenly increase (Li and Born, 2019; Prange et al., 2023; Klockmann et al., 2020).

For the reasons listed above, variability in SPG circulation and convection strength is of considerable interest. Model simulations by Born and Mignot (2012) have shown that the SPG circulation is variable with a period of 15-20 years. They suggested that these internal variations are caused by an advection-convection feedback. In this feedback, strong SPG circulation leads to

lateral advection of salt from the North Atlantic current to the SPG region. However, it takes time for this peak of high salinity to enter the SPG's core. In the meantime, the strong circulation advects salt out of the SPG's centre, leading to lower densities and a decrease in circulation. Once the salinity peak reaches the gyre's center, it facilitates deep convection, which in turn helps to spin up the SPG circulation again. In addition to this advection-convection feedback, SPG behavior has also been to shown to be affected by sea ice cover due to the gyre's sensitivity to buoyancy and wind forcings on longer timescales (Steinsland

et al., 2023)

Besides varying in strength, convection in the subpolar gyre has been observed to shut down altogether. These events are often associated with so-called Great Salinity Anomalies (GSAs). First documented in the late 1970s and early 1980s (Lazier, 1980; Dickson et al., 1988) and observed again in the late 1980s (Belkin et al., 1998) and 2010s (Biló et al., 2022), during these events watermasses of relatively low salinity are advected throughout the Northern seas, sharply decreasing the sea surface

salinity in the region. These fresh water masses have been linked to anomalous sea ice melt in the high Arctic and Nordic seas (Yashayev, 2024; Allan and Allan, 2024). Convection in the Labrador sea can be inhibited temporarily by the passage of GSAs, although anomalous atmospheric conditions and the North Atlantic Oscillation state have also been identified as major forcing mechanisms that can both stop and restart convection (e.g. Gelderloos et al. (2012); Kim et al. (2021); Yashayev (2024)). These observations are in agreement with theoretical results that show, based on the geometry of the basin and a typical atmospheric

forcing, that the Labrador sea is not far from a state that cannot support deep convection (Spall, 2012).

Despite its high variability and importance in linking various elements of the climate system, it is uncertain how the convection in the SPG will respond to climate change. Sgubin et al. (2017) showed that the subpolar gyre region can experience sudden cooling during the 21st century and subsequent reduced temperatures until the end of the simulation. In most models this cooling is caused by a permanent collapse of convection in the SPG. Such an abrupt cooling occurred in 45.5% of the

subset of best performing models in the CMIP5 ensemble, and in 36.4% of the best performing models in CMIP6 (Swingedouw et al., 2021). In addition, analysis of the CESM Large Ensemble has shown that shutdown of deep convection in the North Atlantic can be stochastically triggered, making it hard to predict when this shutdown could occur (Gu et al., 2024). The importance of stochastic atmospheric forcing on triggering instabilities was also noted by Swingedouw et al. (2021).

On the other side of the model spectrum, conceptual models have been used to study the stability of convection and circula-

tion in the North Atlantic subpolar gyre. Most notably, Born and Stocker (2014) formulated a four box model for the convective basin of the Labrador sea. This model builds on Welander (1982), Rahmstorf (2001) and Kuhlbrodt et al. (2001)'s previous conceptual models of convection and extends models of the gyre and circulation in marginal seas by Spall (2004), Straneo (2006), Deshayes et al. (2009) and Spall (2012). In this, a hysteresis in gyre strength for variations in both the gyre current's salinity and freshwater forcing was found. These hystereses are caused by the formation of a strong baroclinic flow when the



density difference between the inside and outside of the gyre is sufficiently high. Although heavily simplified, when forced with reanalyzed surface temperatures the model mostly reproduces the observed gyre strenghts (Born et al., 2015), giving the model credibility.

Models of various complexity indicate that it remains elusive whether the North Atlantic subpolar gyre will continue to support convection in a changing climate. The collapse of convection in the SPG has been recognized as a potential tipping point in the climate system (Loriani et al., 2023). The critical value of global surface temperature associated with this tipping point is estimated to be 1.1 to 3.8 °C of global warming, and changes could occur with a relatively rapid timescale of approximately 10 years (Armstrong McKay et al., 2022). Such a collapse could potentially strongly influence surface temperatures and other climatic variables in the Atlantic and Nordic seas, as well as shifting the atmospheric jet stream (Swingedouw et al., 2021; Loriani et al., 2023). The SPG's connection to the AMOC and the Greenland Ice Sheet is worrisome, as conceptual modelling studies have shown that a freshwater-driven collapse of convection in the gyre could influence the stability of the AMOC (Neff et al., 2023).

In this paper, we investigate the stability of convection in the North Atlantic subpolar gyre using an adjusted version of the simple conceptual model of the SPG that was initially introduced by Born and Stocker (2014). We analyze the bifurcation structure of the model and subsequently apply noise to it, in order to investigate if this noise can induce transitions to a state without convection. Section 2 provides a detailed description of the stochastic model and outlines the methods used. In Sect. 3.1 the bifurcation structure of the model without noise is presented, and various dynamical regimes of this deterministic model are identified in Sect. 3.2. We then study the dynamics of the model when including noise in Sect. 3.3. The results and some limitations of the model are discussed in Sect. 4.

## 2 Model and methods

### 2.1 Model formulation

In this section we present a box model of the SPG. It consists of a center part where convection takes place and a gyre circulation around that center. It is derived from the Born and Stocker (2014) model and in Fig. 1 a diagram of the model is shown. At the center of the model are two vertically stacked cylindrical boxes (box 1, 3), representing the convective basin. These boxes are both surrounded by an annular box (box 2, 4) which represent the surface and deep gyre currents ($U_s$ and $U_d$, respectively) around the center. The ratio of the height of the surface and deep box is denoted by $r$. There are no currents in the inner boxes and convective mixing can only take place between the inner boxes. The four boxes have the following properties:

– Box 1 is the cylindrical inner box at the air-water interface, with temperature $T_1$ and salinity $S_1$. It is the only one of the four boxes in which atmosphere-ocean interactions take place. It is exposed to an atmosphere with temperature $T_0$ and in it the external freshwater flux enters (denoted by $F$). Physically, this freshwater flux consists mostly of precipitation. There is an eddy heat and salt flux ($E$) between box 1 and 2, and convective mixing ($C$) between box 1 and 3 can occur.





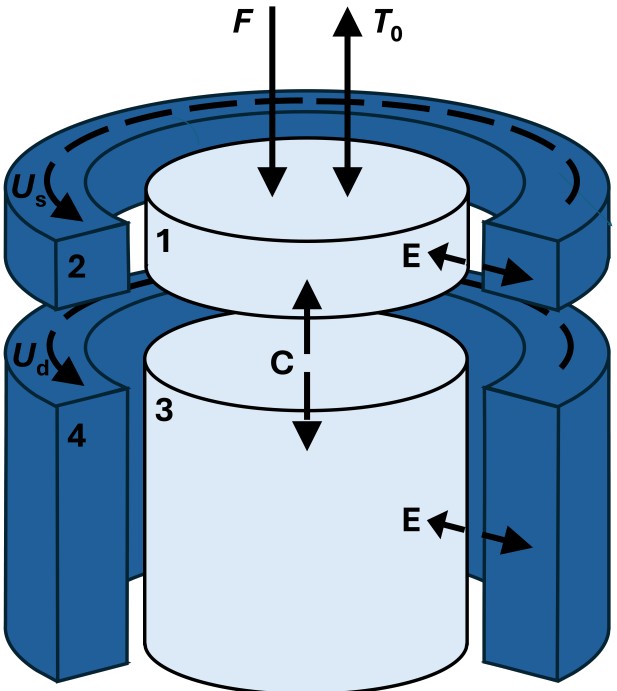

**Figure 1.** Illustration of the four-box model. Numbers 1 to 4 denote the boxes, $U_s$ and $U_d$ the surface and deep gyre current, $F$ the surface freshwater forcing and $T_0$ the atmospheric temperature. Transport between the horizontal boxes can occur as eddy fluxes ($E$). Convection ($C$) can occur between boxes 1 and 3. Figure adapted from Born and Stocker (2014).

- Box 2 is the annular outer box in which the surface gyre current flows, with temperature $T_2$ and salinity $S_2$. Salinity and temperature anomalies that are transported by the gyre current are represented in this box of the model. The upper boundary current is denoted by $U_s$.

- Box 3 is the deep cylindrical inner box, with temperature $T_3$ and salinity $S_3$. There is an eddy heat and salt flux ($E$) between box 3 and 4, and convective mixing ($C$) between box 3 and box 1 can occur.

- Box 4 is the annular outer box in which the deep gyre current flows, with temperature $T_4$ and salinity $S_4$. The lower boundary current is denoted by $U_d$.

In order to perform a bifurcation analysis, we adjusted Born and Stocker (2014)'s original model by non-dimensionalizing it and parameterizing convection continously. In addition, the model was made autonomous by considering a constant, rather than a seasonally varying, atmospheric temperature $T_0$. Lastly, we modelled the effect of noise in the salinity budget by adding additive noise terms in the differential equations that describe the surface box salinity $S_1$ and the surface velocity $U_s$. These noise terms represent stochastic variations in the surface current's salinity $S_2$ and the freshwater flux $F$. Details on the derivation of this model and its relation to the original model can be found in Appendix A.



The adjusted, stochastic model is outlined in Equations (1):

$$U_{\mathrm{d}} = U_{\mathrm{btp}} - \eta[(S_4 - T_4) - (S_3 - T_3)], \tag{1a}$$

$$U_{\mathrm{s}} = U_{\mathrm{btp}} - \eta[(S_4 - T_4) - (S_3 - T_3)] - \eta r[(S_2 + \zeta_S - T_2) - (S_1 - T_1)], \tag{1b}$$

$$\frac{\mathrm{d}T_1}{\mathrm{d}t} = \mu_{\mathrm{H}} U_{\mathrm{s}}(T_2 - T_1) + \mu_{\mathrm{C}}(T_3 - T_1)\mathcal{H}((S_1 - T_1) - (S_3 - T_3)) + \mu_{\mathrm{A}}(T_0 - T_1), \tag{1c}$$

$$\frac{\mathrm{d}S_1}{\mathrm{d}t} = \mu_{\mathrm{H}} U_{\mathrm{s}}(S_2 + \zeta_S - S_1) + \mu_{\mathrm{C}}(S_3 - S_1)\mathcal{H}((S_1 - T_1) - (S_3 - T_3)) - (F + \zeta_F), \tag{1d}$$

$$\frac{\mathrm{d}T_3}{\mathrm{d}t} = \mu_{\mathrm{H}} U_{\mathrm{d}}(T_4 - T_3) - \mu_{\mathrm{C}} r(T_3 - T_1)\mathcal{H}((S_1 - T_1) - (S_3 - T_3)), \tag{1e}$$

$$\frac{\mathrm{d}S_3}{\mathrm{d}t} = \mu_{\mathrm{H}} U_{\mathrm{d}}(S_4 - S_3) - \mu_{\mathrm{C}} r(S_3 - S_1)\mathcal{H}((S_1 - T_1) - (S_3 - T_3)), \tag{1f}$$

$$\frac{\mathrm{d}\zeta_S}{\mathrm{d}t} = -\frac{\zeta_S}{\tau_S} + \sigma_S \xi, \tag{1g}$$

$$\frac{\mathrm{d}\zeta_F}{\mathrm{d}t} = -\frac{\zeta_F}{\tau_F} + \sigma_F \xi, \tag{1h}$$

where $\mathcal{H}(x)$ represents the step-like function

$$\mathcal{H}(x) = \frac{1}{2} + \frac{1}{2}\tanh(kx) \tag{2}$$

with $k \gg 1$. The term $S_1 - T_1$ (and similar for the other boxes) in (1) is a nondimensional expression for the density. Lastly, the dimensionless horizontal volume transport of the gyre is defined as

$$M = rU_{\mathrm{s}} + U_{\mathrm{d}}. \tag{3}$$

The value of this transport is used to distinguish between the gyre's different flow regimes.

The model (1) consists of two velocity terms and six differential equations: four describing the evolution of the prognostic variables $T_1$, $S_1$, $T_3$, $S_3$, and two describing the evolution of the noise terms $\zeta_S$ and $\zeta_F$. All other parameters are prescribed. The expressions, physical interpretations and values of these parameters can be found in Table 1.

The stochastic components of the model, $\zeta_S$ and $\zeta_F$, are Ornstein-Uhlenbeck processes (e.g. Penland and Ewald, 2008; Boers et al., 2022; Ditlevsen and Johnsen, 2010). The evolution of these terms are described by their own differential equations (Equations 1g and h). They consist of a noise term $\sigma_X \xi$, where $\sigma_X^2$ is the variance of the noise and $\xi$ represents a white noise process. In addition, the processes have a relaxation timescale $\tau_X$. The addition of this timescale ensures that the noise is correlated in time and not completely re-initialized at every timestep. This formulation is similar to that in e.g. Dijkstra et al. (2023). The deterministic version of the model, which is discussed in Sects. 3.1 and 3.2, is a special case of (1) with $\zeta_S$, $\sigma_S$, $\zeta_F$ and $\sigma_F$ set to zero.

The noise in (1) is determined by the noise variance $\sigma_X$ and correlation time scale $\tau_X$. When studying the effect of noise on the system we vary the strength up to $\sigma_S = 0.5$ psu and $\sigma_F = 1$ m yr$^{-1}$ to compensate for the lack of seasonality in the model. To some extent, these large amplitudes can then simulate the effect of an anomalously wet or dry season on the transport of





**Table 1.** Prescribed parameters of the model (1). The values were calculated from the default model parameters as outlined in Table 1 of Born and Stocker (2014) and Appendix A. No values are given for the parameters $\eta$, $\mu_H$, $\mu_C$, and $\mu_A$, as these values do not have an intuitive interpretation.

| Parameter | Physical interpretation | Dimensional value | Nondimensional value |
|---|---|---|---|
| $U_{\text{btp}}$ | strength of barotropic current | $0.133 \text{ m s}^{-1}$ | 1 |
| $S_2$ | salinity in surface gyre box | 35 psu | 0.884 |
| $S_4$ | salinity in deep gyre box | 34.9 psu | 0.881 |
| $T_2$ | temperature in surface gyre box | $10\,^{\circ}\text{C}$ | 1.02 |
| $T_4$ | temperature in deep gyre box | $4\,^{\circ}\text{C}$ | 1 |
| $r$ | ratio of surface and deep box height | $7.14 \times 10^{-2}$ | $7.14 \times 10^{-2}$ |
| $\eta$ | strength of thermal wind in deep box | - | $1.29 \times 10^{2}$ |
| $\mu_H$ | horizontal mixing efficiency | - | $8.38 \times 10^{-1}$ |
| $\mu_C$ | convection efficiency | - | $3.41 \times 10^{2}$ |
| $\mu_A$ | atmosphere-ocean exchange efficiency | - | $1.21 \times 10^{1}$ |

the gyre. To simulate the different intrinsic time scales of variability in ocean and atmosphere, correlation time scales of $\tau_S = 1$ yr and $\tau_F = 90$ days were used unless specified otherwise. With these time scales, the stochastic variations in gyre salinity $S_2$ (described by $\zeta_S$) can be interpreted as being driven by external variations in for example sea ice cover, and the stochastic

135 variations in freshwater forcing $F$ (described by $\zeta_F$) as quasi-seasonal variations in precipitation.

It is worth noting that the maximum values of the noise amplitude $\sigma_S$ and $\sigma_F$ were chosen to be quite high. Of course, in reality the salinity of the gyre does not vary with an amplitude of 0.5 psu, and neither does the freshwater forcing vary with an amplitude of $1$ m yr$^{-1}$. These high values were taken to gain a mechanistic understanding of the processes at play, which is difficult when the noise strength is low and does not have much effect. As the model is highly conceptual, these variables

140 should not be interpreted as realistic values of intrinsic noise in the climate system.

## 2.2 Methods

We use two versions of the model (1) in our analysis, as outlined above. When performing the bifurcation analysis (Sects. 3.1 and 3.2), we use the deterministic system. The stochastic version of (1) is used in the simulations where noise is applied to the system (Sect. 3.3).

145 ### 2.2.1 Bifurcation analysis

The Julia continuation software BifurcationKit.jl (Veltz, 2020) was used to calculate bifurcation diagrams of the deterministic version of the system (1) in Sect. 3.1. The standard pseudo-arclength continuation (PALC) algorithm was used with Newton tolerances ranging from $10^{-9}$ to $10^{-11}$, a minimum arclength of $10^{-10}$ and a maximum arclength ranging from $10^{-5}$ to $10^{-6}$.





These values were varied slightly between continuations to ensure that the resulting diagrams were all in qualitative agreement about the structure of equilibria and bifurcations. Continuations were performed with $S_2$, $F$ and $T_2$ as control parameter. In all results shown, the nondimensional model was used for calculations. After analysis units were converted back to dimensional units for ease of interpretation.

Furthermore, in order to determine relevant parameter regimes for further analysis, a two-parameter (codim-2) continuation was performed in Sect. 3.2. Here, the bifurcation points that were found with $S_2$ as control parameter were continued with $F$ as control parameter. This provides an overview of the system's stability landscape as a function of these two parameters.

### 2.2.2 Simulations

The stability of the stochastic system (1) was studied by time integration. These integrations were performed with the Julia library DifferentialEquations.jl (Rackauckas and Nie, 2017a, b). The built-in Euler-Maruyama method for stochastic differential equations was used with 3650 timesteps per year and the model was given 10 years of spin-up time. After this period the model was run for 5000 years unless specified otherwise. The values of the gyre transport $M$ (3) were averaged for every model year and only these average yearly values were used in further analysis.

To quantify how long shutdowns of convection last, residence times in the non-convective state were calculated. As a non-convective state is associated with low gyre transport, the gyre was considered to be in a non-convective state when its yearly averaged transport $M$ was less than 22 Sv for at least one year. When this was the case, the years that the gyre was in this state before transitioning back to a convective state were counted. Kernel densities of the full time series and the residence times were estimated with KernelDensity.jl (Byrne, 2024) to visualize the long term behavior of the model. Lastly, the time the gyre spent in convective and non-convective states was quantified by the state ratio $R$, defined as

$$R = \frac{n_{\text{convection}}}{n_{\text{total}}}, \tag{4}$$

where $n_{\text{convection}}$ is the number of years the gyre is in a convective state with high transport $M$ and $n_{\text{total}}$ is the total length of the time series (5000 years in most cases). When $R = 1$, the gyre is in a convective (high transport) state for all model years, whereas when $R = 0$ the gyre is in a non-convective (low transport) state for all model years.

## 3 Results

To understand the behavior of the subpolar gyre in the determnistic model, we performed a bifurcation analysis with two different control parameters: the gyre salinity $S_2$ and the freshwater forcing $F$. We vary the gyre salinity because changes in Greenland Ice Sheet runoff and precipitation upstream will affect the salinity in the gyre current. Similarly, changes in precipitation in the SPG region will have an effect on the density of the SPG's convective core. Varying these two control parameters thus allows us to study the effect of changes in the gyre's salinity budget on its circulation.



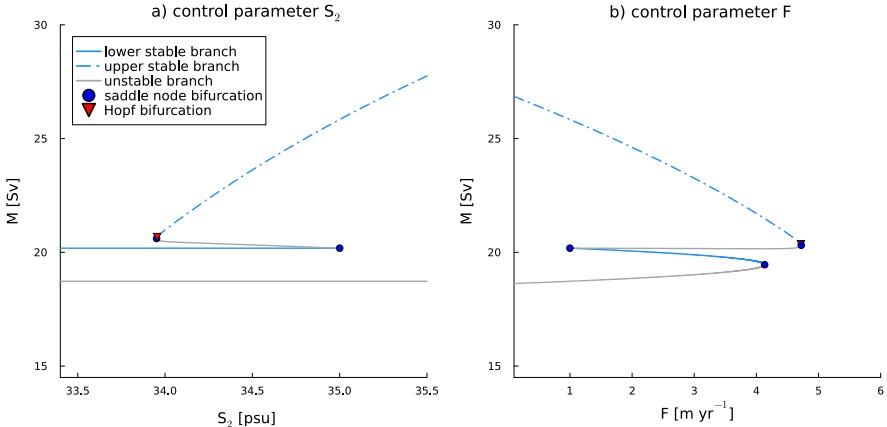

**Figure 2.** Bifurcation diagrams with gyre salinity $S_2$ and freshwater forcing $F$ as control parameter and total gyre transport M on the vertical axis. Bistable regions were found between approximately 34 and 35 psu ($S_2$) and 1 and 4 m yr$^{-1}$ ($F$).

The results of this analysis are presented in Sect. 3.1. Building on these results, the behavior of the deterministic model is categorized into three regimes in Sect. 3.2. Lastly, the behavior of the stochastic model under several noise levels is shown in
Sect. 3.3.

### 3.1 Single-parameter bifurcation analysis

A bifurcation diagram with the gyre salinity $S_2$ as control parameter is shown in Fig. 2a. The solutions are organized into two stable branches, connected by a double fold. For low values of $S_2$ the gyre transport is constant, with $M \approx 20$ Sv. This low-transport stable branch ends in a saddle node bifurcation at $S_2 = 35$ psu. The upper stable branch starts with a Hopf bifurcation,
which is followed very closely by another saddle node. The transport of the upper branch is not constant, but increases in value as $S_2$ increases. An unstable branch connects the two stable branches.

The sensitivity of the gyre to the freshwater forcing $F$ is shown in the bifurcation diagram in Fig. 2b. As in Fig. 2a, a double fold can be seen, consisting of two stable branches connected by an unstable branch. The lower stable branch terminates at a value of $F = 1.00$ m yr$^{-1}$ on the left end and at $F = 4.13$ m yr$^{-1}$ on the right. The higher stable branch terminates in a
Hopf bifurcation which is accompanied by a saddle node bifurcation on the unstable branch very close to the Hopf bifurcation, similar to what was found with $S_2$ as control parameter. Bifurcation diagrams with $T_2$ as control parameter were also computed, but no bistabilies in a relevant parameter regime were found.

In our analysis, we found a Hopf bifurcation very close to a saddle node bifurcation for a wide range of parameters. However, attempts at finding the corresponding periodic orbits were unsuccessful and it is not clear if these Hopf bifurcations are real
characteristics of the model or rather artefacts of the used continuation software. Consequently, these points are shown in the results, but will not be discussed further.





The two stable branches that appear in the bifurcation diagrams (Fig. 2) represent different modes of gyre behavior in the model: the upper branch consists of solutions with convection, whereas the lower branch consists of solutions without convection. This can be deduced from the density differences between the different boxes of the model for the different branches 
(shown in Supplementary Fig. B1 with $S_2$ as control parameter). When the vertical density difference $\sigma_1 - \sigma_3$ is zero, which is the case for the upper branch, there is continuous overturning between boxes 1 and 3. This leads to large horizontal density differences and hence a strong baroclinic contribution to the gyre transport. In this convective regime the four-box model is effectively reduced to three boxes.

When the vertical density difference $\sigma_1 - \sigma_3$ is negative, as is the case in the lower branch, the vertical boxes are stably stratified and no convection can occur. The lower cylindrical box is therefore fully connected to the lower annular box and the density difference between the two deep boxes is equal to zero. Consequently, there is no baroclinic contribution to the deep current $U_d$ and the strength of the gyre is fully determined by the density difference between the upper two boxes and the barotropic current. Essentially, in this non-convective regime the four-box model is reduced to a two-box model.

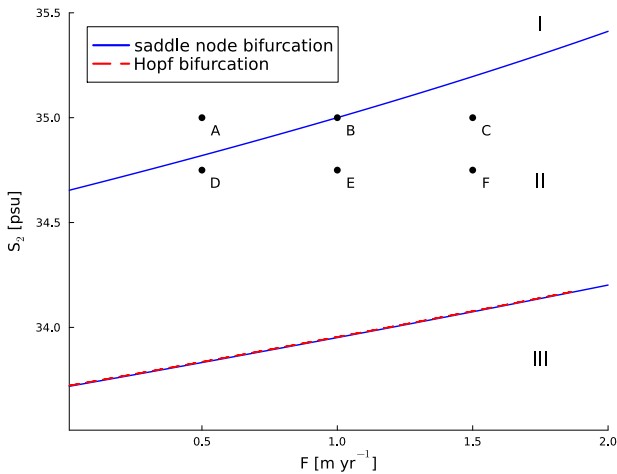

**Figure 3.** Two-parameter bifurcation diagram in the parameter range $0 < F < 2$ m yr$^{-1}$, $33.5 < S_2 < 35.5$ psu. The three dynamical regimes are denoted by roman numerals I-III: regime I is monostable and convective, regime II is bistable, and regime III is monostable and non-convective. Points A to F are points in $(S_2, F)$-space that were used as parameter values in the time integrations performed in Sect. 3.3.

## 3.2 Model regimes

In order to understand how changes in both gyre salinity $S_2$ and freshwater forcing $F$ affect the stability of the gyre, a two-parameter continuation was performed. The two saddle nodes and one Hopf bifurcation were continued in a relevant parameter regime, which is defined here as a regime with $0 < F < 2$ m yr$^{-1}$ and $33.5 < S_2 < 35.5$ psu.

This two-parameter diagram (Fig. 3) shows for which parameter values certain types of behavior can be expected. The lines in this figure represent the location of the bifurcation points, rather than the equilibrium solutions. If a vertical transect at a



**Table 2.** State ratio $R$ (Equation (4)) for different noise levels and parameter values in $(S_2, F)$-space, as defined in Fig. 3. Point A corresponds to $S_2 = 35$ psu, $F = 0.5$ m yr$^{-1}$; point B to $S_2 = 35$ psu, $F = 1$ m yr$^{-1}$; point C to $S_2 = 35$ psu, $F = 1.5$ m yr$^{-1}$; point D to $S_2 = 34.75$ psu, $F = 0.5$ m yr$^{-1}$; point E to $S_2 = 34.75$ psu, $F = 1$ m yr$^{-1}$; and point F to $S_2 = 34.75$ psu, $F = 1.5$ m yr$^{-1}$.

|  | A | B | C | D | E | F |
|---|---|---|---|---|---|---|
| $\sigma_S = 0.25$ psu | 1 | 1 | 1 | 1 | 1 | 0.1268 |
| $\sigma_S = 0.5$ psu | 0.9780 | 0.9366 | 0.7598 | 0.8274 | 0.5354 | 0.2018 |
| $\sigma_F = 0.5$ m yr$^{-1}$ | 1 | 1 | 1 | 1 | 1 | 1 |
| $\sigma_F = 1$ m yr$^{-1}$ ($\tau_F = 90$ days) | 1 | 1 | 1 | 1 | 1 | 1 |
| $\sigma_F = 1$ m yr$^{-1}$ ($\tau_F = 1$ yr) | 1 | 1 | 1 | 1 | 1 | 0.9110 |

given value of $F$ is followed for increasing values of $S_2$, first a saddle-node and Hopf bifurcation are encountered. For higher values of $S_2$ another saddle-node is found. Above this last saddle-node there are no bifurcation points in this parameter range. Such transects with constant $F$ and varying $S_2$ were shown previously in Fig. 2a (for $F = 1$ m yr$^{-1}$). The vertical region between the blue lines representing saddle-node bifurcation in Fig. 3 corresponds to the bistable region in Fig. 2a. Similarly, horizontal transects in Fig. 3 are an extension of the results shown in Fig. 2b (for $S_2 = 35$ psu), but for realistic values of $F$

only one saddle node is encountered.

Based on Fig. 3, three different model regimes can be distinguished. The first regime (denoted I) is monostable and convective. The gyre model is only in this regime for sufficiently high values of $S_2$. Solutions in this regime correspond to high values of gyre transport $M$. In the other monostable regime (III) there is again only one stable solution, but this solution does not have any convection. This corresponds to low values of the gyre transport $M$. In the intermediate regime (II) there are two

stable solutions, one with and one without convection.

According to the reference values that were used in Born and Stocker (2014), the subpolar gyre's current state is on point B in Fig. 3 ($F = 1$ m yr$^{-1}$, $S_2 = 35$ psu), within the bistable regime II. It is therefore potentially susceptible to changes in precipitation, hydrography and circulation, as changes in $F$ or $S_2$ may lead to a transition to a non-convective state. In other words, the gyre can exhibit tipping behavior.

## 3.3 Noise-induced transitions

To study if noise can indeed induce transitions between convective and non-convective states, six different sets of parameter values in $(S_2, F)$-space were selected, as indicated in Fig. 3 with letters A-F. The stochastic model (1) was then time-integrated for long run times to study the effect of noise for different values in parameter space. The six points A-F were chosen to be in a region in $(S_2, F)$-space that is the boundary between regimes I and II, such that points in both the monostable convective

and the bistable convective regime are considered. Within our model context, the present day subpolar gyre region has a gyre salinity $S_2$ of 35 psu and freshwater forcing $F$ of 1 m yr$^{-1}$ (Born and Stocker, 2014). Point B in Fig. 3, representing these present-day values, is defined as the reference forcing.





The other five points hence span variability around present-day values of the parameters, as well as possible deviations from the reference forcing. Point A ($S_2 = 35$ psu, $F = 0.5$ m yr$^{-1}$) represents a reference salinity forcing and a lower-than-reference freshwater forcing (i.e. precipitation). Point C ($S_2 = 35$ psu, $F = 1.5$ m yr$^{-1}$) represents the same salinity forcing as A and B and a higher-than-reference freshwater forcing. Points D, E and F all have a lower-than-reference salinity and the same increasing freshwater forcing as points A, B and C.

The main results from the integrations with noise applied to the gyre salinity ($\sigma_S = 0.5$ psu) are summarized in Fig. 4. The final 500 years of the time series are shown, as well as kernel density estimations of the total time series and the time spent in the non-convective state. In addition, values of the state ratio $R$ (Equation (4)) for integration with $\sigma_S = 0.25$ psu (not shown in Fig.) and $\sigma_S = 0.5$ psu are shown in Table 2 as a measure of how much time the gyre spends in a convective state. Clearly, the integrations for the six parameter sets A-F exhibit different behavior. Changing the values of the parameters in ($S_2, F$)-space can have profound influences on the dynamics of the gyre.

A first observation is that increasing the freshwater forcing $F$ always leads to less convection in the gyre. For parameter set A, the gyre is almost never in a state without convection, as can be seen in Fig. 4b. As $F$ increases, the amount of years spent in the non-convective state increases. This can be seen in the small bump in the kernel density of the gyre transport around 20 Sv for parameter set B (Fig. 4e) and much more clearly for parameter set C (Fig. 4h). Similar behavior can be observed in the progression from D to F. The gyre spends most of its time in the non-convective state for parameter set F (Fig. 4q and R $< 0.5$, Table 2).

Similarly, decreasing the background salinity $S_2$ always leads to an increase in the amount of years without convection. When comparing cases with the same freshwater forcing but different salinity (A versus D, B versus E, C versus F), the gyre always spends fewer years in the convective state for the case with lower salinity. Taken together, increasing $F$ and decreasing $S_2$ destabilizes the gyre's convection, whereas decreasing $F$ and increasing $S_2$ stabilizes it. The opposing sign of these two effects is not surprising, as both decreasing $S_2$ and increasing $F$ reduces salinity levels in the gyre.

These destabilizing factors increase the total amount of years spent in the non-convective state, as well as increasing the average time a non-convective period lasts. The latter is shown in the rightmost column of Fig. 4. When the gyre is stable and spends most of its time in a convective state, excursions to the non-convective state are short, typically lasting much less than 10 years. This can be seen in, for example, cases A (Fig. 4c) and D (Fig. 4i). As the gyre's convection destabilizes, the kernel density estimate of time spent in the non-convectives state broadens to longer times. The peak also shifts to longer residence times. This is most visible for case F (Fig. 4r) and to a lesser extent also for cases C and E (Figs. 4i and o).

It is worth noting that even in the least stable case studied here (F), the gyre never becomes fully non-convective. The length of the non-convective periods increases, and most of the time is spent in the non-convective state, but there are still many convective episodes. Qualitatively, the gyre transitions from a convective (high transport) state with non-convective (low transport) episodes to a non-convective (low transport) state with convective (high transport) episodes. In the time series (Fig. 4p) these episodes appear as 'excitations' from a more stable low transport mean state. However, both branches are still stable solutions (Sect. 3.1).



These results, where the amplitude of the applied noise is $\sigma_S = 0.5$ psu, are summarized in the second row of Table 2. The destabilization of the gyre's convection is represented here by a decrease in state ratio $R$. Interestingly, for this noise level, convection ceases in all cases studied at least for some years. Even case A, for which the deterministic parameter values are

clearly in the monostable convective regime (Fig. 3), has rare events where the convection collapses. This is not completely unexpected, as the applied noise amplitude (0.5 psu) is greater than the distance from the regime boundary (approximately 0.2 psu). It does, however, show that being in a 'safe' monostable parameter regime does not guarantee that convection in the gyre will never destabilize under stochastic forcing.

Conversely, when the parameter values in $(S_2, F)$-space are far in the bistable regime, a low noise level can result in few

transitions and therefore a strongly non-convective gyre. This is the case for parameter set F when $\sigma_S = 0.25$ psu (Table 2): here, because the applied noise level is low, transitions between states are rare and the gyre can end up spending most of its time in a non-convective state in a time integration. This phenomenon seems to be governed mostly by chance and not by the stability of the system, as the basin of attraction of the convective branch is much bigger than that of the non-convective branch (Fig. 2).

Both the gyre salinity $S_2$ and the freshwater forcing $F$ can exhibit variability. To study the effect of the latter, the model was also integrated with added noise in the freshwater forcing instead of noise in the gyre salinity. The state ratios $R$ from these integrations ($\sigma_F = 1$ m yr$^{-1}$) are summarized in Table 2. These results show a drastically different picture than those discussed above: for virtually all of the performed simulations, the gyre always remains in a convective state with high transport.

This is somewhat surprising. The noise amplitude $\sigma_F = 1$ m yr$^{-1}$ is twice that of the noise applied to $S_2$ shown in Fig. 4,

yet no transitions between the convective and non-convective state are observed when the timescale $\tau_F$ of the noise in $F$ is 90 days. Even when $\tau_F$ is increased to one year, the gyre only stops convecting a few times in case F (Table 2). In this scenario, the gyre again only has non-convective episodes for a high value of $F$ and a low value of $S_2$. This is in line with the results shown above, in which convection in the gyre was the least stable for case F. These simulations indicate that the convection in the SPG is less sensitive to noise in the freshwater forcing than to noise in the background gyre salinity.

In all results thus far presented, noise was only applied in either the gyre salinity or the freshwater term. Adding noise in both terms simultaneously could have a different effect on the gyre's behavior. To this end, integrations were performed with various combinations of the noise amplitudes $\sigma_S$ and $\sigma_F$. The results of these integrations are shown in Fig. 5. Clearly, $\sigma_S$ is the main driver of noise-driven transitions. The rightmost columns have the lowest value of $R$ for cases B, C, D, E, and F. In other words, as the noise in $S_2$ increases, the gyre spends more time in a non-convective state.

Contrastingly, increasing the noise in the freshwater forcing does not seem to influence the behavior of the gyre much. For cases A, B, and C, there is no clear decrease in $R$ as $\sigma_F$ increases. For the other cases, large noise amplitudes in the freshwater forcing slightly increase the amount of time spent in the non-convective state. This can be seen in the hatched cells corresponding to high values of $\sigma_F$ for parameter set D (column of $\sigma_S = 0.4$ psu), E ($\sigma_S = 0.3$ psu) and F ($\sigma_S = 0.2$ psu). A possible explanation for this is that these parameter sets are far in the bistable regime and transitions between states are

easier here. The additional forcing of $\sigma_F$ can help induce a transition to a non-convective state, but only when $\sigma_S$ is sufficiently high. In other words, high noise in the freshwater forcing can lower the threshold of noise in salinity that is required to stop





convection in some cases. Overall, applying noise to the gyre salinity has a much greater effect than applying it to the freshwater forcing. This low sensitivity to noise in the freshwater forcing will be discussed extensively in Sect. 4.2.

## 4 Discussion and Conclusions

We used a conceptual model of the SPG to investigate the occurrence of periods with and without convection, and how the presence of noise in salinity and precipitation influence the stability of convection. In this section, we summarize the results from the bifurcation analysis in Sect. 4.1. This is followed by a discussion of the model's sensitivity to noise in the gyre salinity in Sect. 4.2. Some limitations of the used model are outlined in Sect. 4.3 and lastly, the results of this article are put into a broader context in Sect. 4.4.

### 4.1 Bifurcation structure

In Sect. 3.1 it was shown that there is a bistability in the gyre's transport $M$ for certain (combinations of) values of $S_2$ and $F$. This is in agreement with the results found in Born and Stocker (2014). The main difference between our results and those in Born and Stocker (2014) is the width of the hysteresis: in our results, bistabilities are present for a larger range of parameter values. This is caused by the absence of a seasonal cycle in the forcing of the model (1) and is discussed further in Sect. 4.3.

320  The bifurcation structure of the bistability is a double fold, and the two branches of these folds exhibit very different behavior. The high transport ($M$) branch corresponds to a gyre with permanent convection, whereas the low transport branch corresponds to a gyre with no convection. Based on the parameter values in ($S_2, F$)-space, in Sect. 3.2 we distinguished three different regimes for the gyre's dynamical state: a monostable convective, a bistable, and a monostable non-convective regime. In and close to this bistable regime, noise can induce transitions from one state to the other. No region of bistability was found when

using $T_2$ as the control parameter. The SPG is therefore, at least in this model, much more sensitive to haline pertubations in the surface gyre boxes than to temperature pertubations.

### 4.2 Strong sensitivity to noise in gyre salinity

In Sect. 3.3, we studied the effect of noise on the SPG's behavior. We found that convection in the gyre can temporarily collapse due to stochastic forcing. The likelihood and severity of these collapses is strongly influenced by the choice of parameters in

($S_2, F$)-space. Decreasing the surface current salinity $S_2$ and increasing the freshwater forcing $F$ both destabilize the SPG and increase its sensitivity to noise-induced collapses of convection. As the gyre destabilizes, a state without convection occurs more often and the time that is spent in each non-convective period increases. However, even in the least stable parameter regime studied here convection in the gyre never fully collapses, and it is always possible to restart convection stochastically. Similarly, when the gyre is in a monostable convective regime, convection can still ocassionally collapse and restart stochasti-

cally.

 A striking result is that noise in the freshwater forcing $F$ does not have nearly as much of an influence on the SPG's behavior as noise in the gyre salinity $S_2$. Transitions are almost exclusively driven by noise in the salinity. A possible reason for this



result is that the noise terms are applied to different boxes of the conceptual model: the noise in $F$ is applied to the inner surface box 1, whereas the noise in $S_2$ is applied to the surface current box 2. The stochastic freshwater forcing $\zeta_F$ is therefore distributed over a larger volume than the stochastic salinity forcing $\zeta_S$ and the effective applied concentration is lower. This could partially explain the relatively small effect of adding noise in $F$. However, as salinity is not conserved in this model, it is hard to quantify the effect of these different volumes. Another possible explanation could be that the dynamics of noise added to $S_2$ are more involved and nonlinear than those of $F$. In the model (1), the freshwater term $-F + \zeta_F$ is simply added to the rest of the terms, whereas the salinity term $S_2 + \zeta_S$ is multiplied by the surface current strength $U_s$, making it a strongly nonlinear term. It seems likely that the difference in impact of adding noise in $S_2$ and $F$ is, at least in part, caused by the nonlinear nature of the baroclinic transport in this model.

The sensitivity to noise in the gyre salinity can be interpreted in the context of the Great Salinity Anomalies introduced in the introduction (Gelderloos et al., 2012; Kim et al., 2021; Yashayev, 2024). The passage of such an anomaly through the gyre can be seen as a stochastic deviation from the gyre's mean salinity. The modelled gyre's response to these anomalies by switching from a convective to a non-convective state is therefore in agreement with the observed collapse of convection in the gyre during GSAs. It is remarkable that the highly simple and idealized Born and Stocker model can reproduce these results. This is an indication that this model, simple as it is, captures some of the main physical processes that are at play in the SPG, substantiating the results presented in their work and here. It also emphasizes the continued relevance of using simple box models in oceanographic research.

Ultimately, the behavior of the gyre in this simple stochastic model qualitatively and quantitatively matches the observed behavior, despite the noise being on the high end of realistic values (see discussion in Sect. 2.1). For parameter set B, representing current oceanographic conditions, the gyre spends approximately 6% of its time in a non-convective state. Considering the frequency of collapses of convection that were observed in the previous half century (Sect. 1; Lazier, 1980; Dickson et al., 1988; Belkin et al., 1998; Biló et al., 2022) this is a realistic value. For this reason the used values of noise in especially the gyre salinity are not deemed too high, given the simplifications and assumptions in the model itself.

It would have been interesting to also add a noise term in the atmospheric temperature, as anomalous atmospheric conditions have been linked to shutdowns of convection in the Labrador sea (Gelderloos et al., 2012; Yashayev, 2024). However, the autonomous model setup used here is unsuitable for this. Because there is no seasonality in the autonomous model, the rate of convection is directly set by the atmospheric temperature $T_0$. Small stochastic deviations from $T_0$ would therefore have a direct and major effect on the rate of convection and the gyre's transport. In the more subtle original Born and Stocker (2014) model with a seasonal cycle, noise in the atmospheric temperature could be interpreted as year-to-year variability. However, the bifurcation analysis could only be performed with an autonomous model and therefore, to keep all results consistent with each other, only the autonomous model was analyzed. In the following subsection we discuss how not including a seasonal cycle in the simulations affects the conclusions that can be drawn from them.



## 4.3 Limitations of the conceptual model

All results in this paper were derived from the adjusted model (Sect. 2.1) of convection in the Labrador Sea. Although conceptual models can provide useful results and ease the interpretation of other model studies, they are highly simplified and their results should be interpreted as such. In addition, all analysis was based on an autonomous version of the model. This model, which does not have a seasonal cycle, might behave quite differently from the non-autonomous original model. Convection in the Labrador sea is highly seasonal and only happens in winter when atmospheric temperatures are sufficiently low. By removing this seasonality the model becomes much less subtle, begging the question how realistically the results found in this article represent convection in the SPG.

In order to quantify how removing the seasonal cycle affects the results, a non-autonomous version of the model with a seasonal cycle was simulated with the same stochastic forcings in $S_2$ as the autonomous model. The results of the simulations with noise applied to the gyre salinity are shown in Supplementary Fig. B2. As before, convection in the gyre becomes less stable as $S_2$ decreases and $F$ increases. In addition, the years that are spent in the convective and non-convective states (second column) are distributed approximately the same for the same parameter sets in the non-autonomous and autonomous states. The main difference between the non-autonomous and automous models is that in the non-autonomous model, the gyre typically spends less time in a non-convective states before returning to a convective state (rightmost column of Supplementary Fig. B2). The seasonal cycle in atmospheric temperature makes conditions favorable for convection in winter and extremely unfavorable in summer. On long timescales, this causes the gyre to spend approximately the same time in both states, but the transitions between states become more frequent. This is reflected in the shorter residence times.

Overall, the results from the autonomous and non-autonomous model integrations are in good qualitative correspondence. This indicates that the underlying bifurcation structure of the two models is similar and substantiates the results presented in this article. The fact that such a simple model can replicate collapse and restart of convection in the gyre with simple stochastic forcing is an indication that this is a fundamental mode of SPG behavior.

Only the impact of changes in parameter values on the SPG's behavior was studied here. These are of course not the only factors that can change the stability of convection. For example, one can imagine that the ratio between the depths of the two boxes has some influence on the stability of convection in the gyre. This was investigated in some detail by Born and Stocker (2014), but it would be interesting to formalize their analysis and use continuation software to systematically study the importance of geometric effects. A simple model with variable convection depth would be a much more realistic description of year-to-year convection in the Labrador sea, allowing for an analysis of the effects of preconditioning of the water column between years.

## 4.4 Resilience of the SPG

Based on the results presented here, it can be concluded that convection in the North Atlantic subpolar gyre is quite stable under current oceanographic conditions in a simple model. Convection may occasionally collapse due to stochastic forcing but is always restored again. As the gyre's salinity decreases and the freshwater forcing increases, convection in the gyre collapses



more often and non-convective episodes last longer. However, in none of the studied parameter sets with different salinity and freshwater forcing does convection in the gyre stop altogether. Restart of convection is always possible.

This resilient quality of the SPG was not found in previous model studies with CMIP5 and CMIP6 models (Swingedouw et al., 2021; Sgubin et al., 2017). In most of these models the collapse of convection in the SPG is permanent during the relatively short run time of the simulations. This discrepancy between the conceptual model results and Earth System Model (ESM) results could be an indication that the Born and Stocker model is an overly simplistic representation of the gyre, for example due to the model limitations outlined above. Further research could look at whether some fundamental destabilizing

process is missing in this conceptual model.

The other possibility is that collapse of convection is a much more frequent process than currently thought. This might be due to the relatively low availability of data from high complexity models and observations. Simulation of the climate with ESMs is time-consuming and expensive, and it is not feasible to obtain long time series of climate in the SPG region. It might be the case that convection spontaneously restarts when these models are run for a longer time. Alternatively, it is possible that

CMIP5 and CMIP6 models do not capture the stochastic nature of convection in the Labrador Sea. Even in our current climate, convection in this region is not permanent but sometimes ceases and restarts during Great Salinity Anomalies and anomalous atmospheric conditions. Capturing complex air-ice-sea interactions is complicated, and it is possible that these processes are not resolved fully in ESMs.

If the results here are taken at face value, collapse of convection does not have to be permanent. It would be interesting to

know what effect the switching between states of convection and no convection in the Labrador sea has on the stability of the AMOC. In AMOC stability studies, it is often assumed that the collapse of convection in one of the AMOC's convective basins is relatively permanent and a precursor to other changes (e.g. Neff et al. (2023) and references therein). The consequences of such a permanent collapse on AMOC stability are presumably different than when convection collapses and restarts again on a timescale that is fast for the AMOC. In addition, the relation between the SPG and the Greenland Ice Sheet and Arctic sea

ice should be studied in more detail, as these systems provide much of the low-salinity water that can potentially destabilize convection in the gyre (Dukhovskoy et al., 2021; Malles et al., 2025).

In conclusion, further research is needed to conclusively determine how stable convection in the North Atlantic subpolar gyre really is. Moreover, more insight is needed into whether or not convection in ESMs can restart (or move to different regions), and how this relates to the AMOC strength. The interpretation of the collapse of convection as either a tipping point

in the climate system or as a relatively common process induced by the stochastic nature of oceanic and atmospheric forcing continues to hold important scientific and societal relevance.

*Code availability.* The code used for producing the results is available at https://zenodo.org/doi/10.5281/zenodo.15332921 (van der Heijden, 2025).



**Table A1.** Prescribed parameters of dimensional model (A2). The typical values are identical to the ones outlined in Table 1 of Born and Stocker (2014).

| Parameter | Physical interpretation | Typical value |
|:---:|:---|:---|
| $r$ | radius of the inner box | $3 \times 10^5$ m |
| $w$ | width of the outer box | $1 \times 10^5$ m |
| $h$ | depth of the surface box | $1 \times 10^2$ m |
| $d$ | depth of the deep box | $1.4 \times 10^3$ m |
| $S_0$ | reference salinity | 35 psu |
| $c_1$ | horizontal mixing efficiency | $2.0 \times 10^{-7}$ m$^{-1}$ |
| $c_2$ | convection efficiency | $2.0 \times 10^{-4}$ s$^{-1}$ |
| $\rho_0$ | reference density | 1026 kg m$^{-3}$ |
| $g$ | gravitational acceleration | 9.81 m s$^{-2}$ |
| $f$ | Coriolis parameter | $1.19 \times 10^{-4}$ s$^{-1}$ |
| $\alpha$ | coefficient of thermal expansion | 0.11 kg m$^{-3}$ K$^{-1}$ |
| $\beta$ | coefficient of saline contraction | 0.77 kg m$^{-3}$ |
| $\tau$ | atmosphere-ocean exchange efficiency | $2.6 \times 10^6$ s (30 days) |
| $S_2$ | salinity in surface gyre box | 35 psu |
| $S_4$ | salinity in surface gyre box | 34.9 psu |
| $T_2$ | temperature in surface gyre box | 10 °C |
| $T_4$ | temperature in deep gyre box | 4 °C |
| $T_0$ | atmospheric temperature | 6 °C |
| $T_0^{\mathrm{amp}}$ | amplitude of seasonal cycle in atmospheric temperature | 4 °C |
| $\omega$ | timescale of seasonal cycle | $2\pi$ (365 days)$^{-1}$ |
| $F$ | strenght of freshwater forcing | 1 m yr$^{-1}$ |
| $U_{\mathrm{btp}}$ | strenght of barotropic current | 0.133 m s$^{-1}$ |

## Appendix A: Derivation of the model

The model as described in Born and Stocker (2014) consists of the definition of density

$$\sigma = -\alpha T + \beta S, \tag{A1}$$





and the six coupled equations

$$U_{\mathrm{d}} = U_{\mathrm{btp}} - \frac{gd}{2f\rho_0 w}(\sigma_4 - \sigma_3),$$

$$U_{\mathrm{s}} = U_{\mathrm{d}} \quad - \frac{gd}{2f\rho_0 w}(\sigma_2 - \sigma_1),$$

$$\frac{\mathrm{d}T_1}{\mathrm{d}t} = c_1 U_{\mathrm{s}}(T_2 - T_1) + \tau^{-1}(T_0 - T_0^{\mathrm{amp}}\cos(\omega t) - T_1],$$

$$\frac{\mathrm{d}S_1}{\mathrm{d}t} = c_1 U_{\mathrm{s}}(S_2 - S_1) - F_{\mathrm{s}},$$

$$\frac{\mathrm{d}T_3}{\mathrm{d}t} = c_1 U_{\mathrm{d}}(T_4 - T_3),$$

$$\frac{\mathrm{d}S_3}{\mathrm{d}t} = c_1 U_{\mathrm{d}}(S_4 - S_3).$$

(A2)

The total horizontal volume transport by the flow can be expressed as

$\qquad M = U_{\mathrm{s}} w h + U_{\mathrm{d}} w d.$ (A3)

Convection between box 1 and 3 is not included in the model with a prognostic equation. Instead, at every time step the density anomalies $\sigma_1$ and $\sigma_3$ are compared. When $\sigma_1 > \sigma_3$, the two volumes are mixed by taking the volume-weighted average of the density of the two boxes.

We adjusted several aspects of this model. Firstly, in order to capture the convection process in a formal bifurcation analysis,

it needed to be parameterized without discontinuities in time or parameter space. This was be done by employing a step-like function

$$\mathcal{H}(x) = \frac{1}{2} + \frac{1}{2}\tanh(kx),$$

(A4)

where $k \gg 1$ (e.g. Dijkstra, 2004). In this study a value of $k = 10^5$ was used. The convection between box 1 and 3 was then implemented in the temperature and salinity flux equations, leading to the governing equation

$\qquad \dfrac{\mathrm{d}T_1}{\mathrm{d}t} = c_1 U_{\mathrm{s}}(T_2 - T_1) + \dfrac{d}{d+h}c_2\mathcal{H}(\sigma_1 - \sigma_3)(T_3 - T_1) + \tau^{-1}(T_0 - T_0^{\mathrm{amp}}\cos(\omega t) - T_1),$ (A5)

and similar for the equations describing $S_1$, $T_3$, and $S_3$. Here $c_2$ (with $c_2 \gg c_1$, taken here as $c_2/c_1 = 10^3$ m s$^{-1}$) is the efficiency of mixing due to convection, and $h$ and $d$ are the depth of the upper and lower box, respectively. The convection terms are multiplied by the $d/(d+h)$ and $h/(d+h)$ to account for the different volumes of the boxes and conserve temperature and salinity during the convective mixing. The values of all dimensional parameters are shown in Table A1.

The resulting equations were non-dimensionalized by multiplying the temperatures by $1/T_4$ and the salinities by $\beta/(\alpha T_4)$. This results in the non-dimensional expressions $\hat{T}_1$ and $\hat{S}_1$

$$\hat{T}_1 = \frac{T_1}{T_4},$$

(A6)

$$\hat{S}_1 = \frac{\beta S_1}{\alpha T_4},$$

(A7)



**Table A2.** Relation between nondimensional and dimensional parameters.

| Nondimensional parameter | Physical interpretation | Dimensional form |
|:---:|:---|:---:|
| $\eta$ | strength of thermal wind in deep box | $\dfrac{gd}{2f\rho_0 w}\dfrac{\alpha T_4}{U_{\mathrm{btp}}}$ |
| $r$ | ratio of surface and deep box height | $\dfrac{h}{d}$ |
| $\mu_{\mathrm{H}}$ | horizontal mixing efficiency | $c_1 t_a U_{\mathrm{btp}}$ |
| $\mu_{\mathrm{C}}$ | convection efficiency | $c_2 t_a \dfrac{d}{h+d}$ |
| $\mu_{\mathrm{O}}$ | atmosphere-ocean exchange efficiency | $\dfrac{t_a}{\tau}$ |
| $\mu_{\mathrm{F}}$ | strength of freshwater forcing | $\dfrac{F S_0}{h}\dfrac{t_a \beta}{\alpha T_4}$ |

and similar for the other temperatures and salinities. This scaling in the temperatures requires the original temperatures to be

given in Kelvin. Note that in this form of the equations, $\hat{T}_4$ is always equal to 1.

The nondimensional form of the freshwater forcing is

$$\hat{F} = \frac{F S_0}{h}\frac{t_a \beta}{\alpha T_4} \tag{A8}$$

In addition, the expressions for the velocities are scaled by $U_{\mathrm{btp}}$ such that $\hat{U}_s = U_s/U_{\mathrm{btp}}$ and $\hat{U}_d = U_d/U_{\mathrm{btp}}$. The time is scaled

by $t_a$, where $t_a$ is defined as one year. Lastly, the ratio between the height of the two boxes is introduced as $r = h/d$. Using the

expression for density and omitting the hats results in the following non-dimensional expression of the model:

$$
\begin{aligned}
U_d &= U_{\mathrm{btp}} - \eta[(S_4 - T_4) - (S_3 - T_3)], \\
U_s &= U_{\mathrm{btp}} - \eta[(S_4 - T_4) - (S_3 - T_3)] - \eta r[(S_2 - T_2) - (S_1 - T_1)], \\
\frac{dT_1}{dt} &= \mu_{\mathrm{H}} U_s (T_2 - T_1) + \mu_{\mathrm{C}}(T_3 - T_1) + \mu_{\mathrm{A}}(T_0 - T_0^{\mathrm{amp}}\cos(t) - T_1), \\
\frac{dS_1}{dt} &= \mu_{\mathrm{H}} U_s (S_2 - S_1) + \mu_{\mathrm{C}}(S_3 - S_1) - F, \\
\frac{dT_3}{dt} &= \mu_{\mathrm{H}} U_d (T_4 - T_3) - \mu_{\mathrm{C}} r(T_3 - T_1), \\
\frac{dS_3}{dt} &= \mu_{\mathrm{H}} U_d (S_4 - S_3) - \mu_{\mathrm{C}} r(S_3 - S_1),
\end{aligned}
\tag{A9}
$$

where $r$, $\eta$, $\mu_{\mathrm{H}}$, $\mu_{\mathrm{C}}$ and $\mu_{\mathrm{A}}$ are dimensionless parameters. The expressions, physical interpretations and typical values of

these parameters can be found in Table 1, and their relation to the original dimensional parameters in Table A2. This approach

is similar to that of Rahmstorf (2001), in which the four-box model is also scaled to the deep 'outer' box that acts as a reservoir.

More generally, the scaling of temperature by a reference temperature $1/T_r$ and salinity by a term like $\alpha/\beta T_r$ bears strong

similarity to the approach of e.g. Dijkstra (2024) and Cessi (1994).

Density does not explicitly enter the non-dimensionalized equations. Instead, the non-dimensional difference $S - T$ acts as a

density. This result can be derived by dividing the expression for density (A1) by $\alpha T_4$. The condition under which convection

occurs consequently becomes $(S_1 - T_1) - (S_3 - T_3) > 0$, or $(S_1 - T_1) > (S_3 - T_3)$. This resembles what was found by Rahmstorf

475  (2001).



Lastly, it is possible to define a dimensionless volume transport from (A3). This can be done by dividing $M$ by $wdU_{\text{btp}}$ such that $\hat{M} = M/(wdU_{\text{btp}})$. Dropping the hat once again results in the dimensionless expression for volume transport

$$M = rU_{\text{s}} + U_{\text{d}}. \tag{A10}$$

## Appendix B: Supplementary Figures

*Author contributions.* ASvdH and SKJF designed the research and KJvdH carried it out, developed the model code and performed the simulations. KJvdH prepared the manuscript with contributions from all authors.

*Competing interests.* The authors declare that they have no conflict of interest.

*Acknowledgements.* This publication is part of the project 'Interacting climate tipping elements: When does tipping cause tipping?' (with project number VI.C.202.081 of the NWO Talent programme) financed by the Dutch Research Council (NWO). This is ClimTip contribution
#65; the ClimTip project has received funding from the European Union's Horizon Europe research and innovation programme under grant agreement No. 101137601.



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



**Figure 4.** Time series and statistics of time integrations of the stochastic SPG model with noise level $\sigma_S = 0.5$ psu. The letters A to F on the left denote the sets of parameter values in $(S_2, F)$-space, as defined in Figure 3. Subfigure a (d, g, j, m, p): time series of gyre transport $M$ for the last 500 model years. The gyre is considered to be non-convective when $M < 22$ Sv, indicated by grey shading in the time series. Subfigure b (e, h, k, n, q): kernel density of the gyre transport $M$, showing the relative occurence of years with and without convection. Subfigure c (f, i, l, o, r): kernel density of time spent in the state with no convection. Note that the y-axis of this right column is scaled differently for every plot. All integrations were performed for 5000 model years and yearly averaged values of these 5000 model years were used to calculate the kernel density estimations.




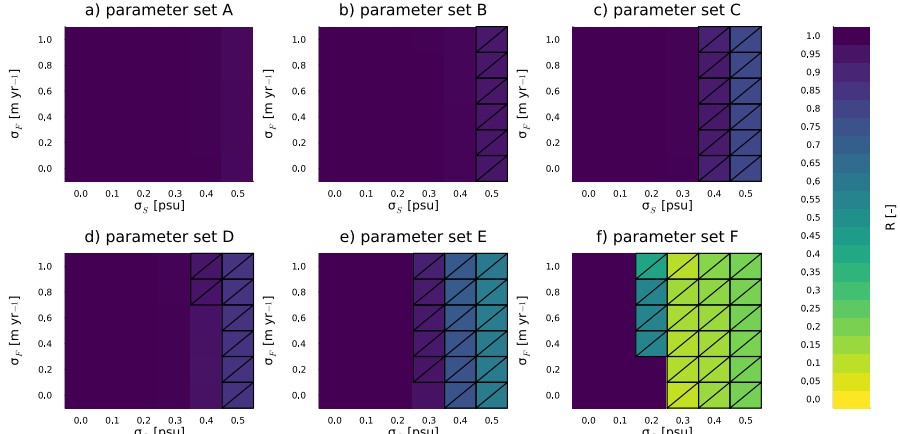

**Figure 5.** State ratio $R$ for different combinations of noise amplitudes $\sigma_S$, $\sigma_F$ for the six parameter sets (subfigures a-f). Each value of $R$ was calculated for every combination from a time integration of 10000 years. Cells are hatched when $R < 0.95$, indicating that the gyre spent more than 5% of the model years in a nonconvective state.

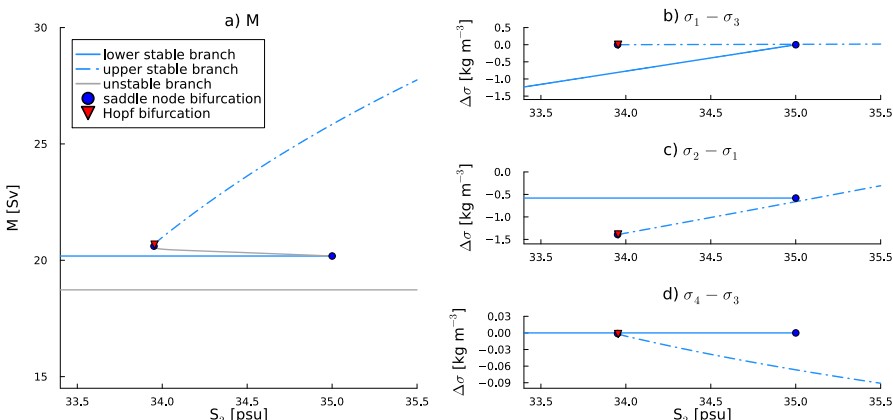

**Figure B1.** The two stable branches in terms of transport $M$ and horizontal and vertical density differences. Subfigure a: bifurcation diagram with $S_2$ as control parameter and $M$ on the vertical axis. This is the same result as shown in Figure 2a. Subfigures b-d: bifurcation diagrams with $S_2$ as control parameter and vertical (subfigure b) and horizontal surface and deep (subfigures c and d, respectively) density differences on the vertical axis.





**Figure B2.** Time series and statistics of time integrations of the non-autonomous model with noise level $\sigma_S = 0.5$ psu. The letters A to F on the left denote the sets of parameter values in $(S_2, F)$-space, as defined in Figure 3. Subfigure a (d, g, j, m, p): time series of gyre transport $M$ for the last 500 model years. Subfigure b (e, h, k, n, q): kernel density of the gyre transport $M$, showing the relative occurence of years with and without convection. Subfigure c (f, i, l, o, r): kernel density of time spent in the state with no convection. Note that the y-axis of this right column is scaled differently for every plot. All integrations were performed for 5000 model years. Yearly averaged values were used to calculate the kernel density estimates.