# Peer review of "The effect of noise on the stability of convection in a conceptual model of the North Atlantic subpolar gyre"

_EGUsphere, 2025_

## Referee Comment (RC1)

egusphere-2025-2074

The effect of noise on the stability of convection in a conceptual model of the North Atlantic subpolar gyre

General Comments

This study uses a conceptual box model of the sub polar gyre to perform a bifurcation analysis and study noise-induced tipping. They find this model to be bi-stable, and test tipping from different initial conditions in phase space. As expected, the initial parametrisation and noise strength contribute to the likelihood of collapse in the time window observed. While an interesting model of the SPG, the model details and the implementation of noise needs to be much better explained and justified. I find that the results presented can be interpreted as simply confirming already known results of noise processes, and may not have such high impact. The grammar, equation, and some figure presentations need to be revised.

Specific Comments

- Section 2.1 I have difficulty understanding how the noise has been added to the model equations. First, it is not clear to me why there are not dynamical equations for boxes 2 and 4- this should be explained. Then, since there is no dynamic equation for box 2, the noise on S_2 is added to Equation 1d. I understand that since the noise is applied as an OU process, these noise processes have different time scales and will represent different physical processes. However, the way the noise is presented in the equations, this still appears to me as two noise processes added to box 1, not one on each box (1 and 2), therefore all noise processes would affect just box 1. The amplitude of the noise term with zeta_S will also be heavily influenced by the pre-factors, and it is not clear to me that this is fully taken into account. Similarly, why would precipitation only affect the surface gyre current and not the surface core box? What does 'precipitation upstream' mean (l.175)?

- Section 3.3 It is not clear to me that the collapses discussed are full tipping events? Are these full transitions to the alternative state? In the discussion lines ~250-270. It is noted 'even in the least stable case... the gyre never becomes fully non-convective'. It is therefore not clear to me that a full tipping event has taken place? What does the alternative stable state correspond to, and therefore is this an expected result? Given the bifurcation diagram is found, could one not check if the system has actually transitioned to the alternative state and is in the alternative basin of attraction?

- The results discussed on page 11, I think are expected mathematically. As you move the initial conditions in phase space, you essentially start the simulations with different effective potential barrier heights for transitions. Therefore, with the same noise amplitude and a fixed time, a different percentage of transitions will take place according to large deviation theory (Freidlin & Wentzell, 1984, Bouchet & Reygner, 2016). Additionally, in some of these 'short excursions' if there is not a full transition to the alternative state, this would then be an example of a noise event followed by a noise-induced recovery (Chapman, Ashwin et al 2024). This could be checked, and the tipping (or not) mechanisms could be identified since the bifurcation diagram is known.

Minor Comments

- Grammar and phrasing needs revising throughout
- Model equations need to be defined more rigorously, not all variables are defined, should be defined immediately after the Equations.
- Where do the values of the model parameters come from? Literature, GCMs, physical estimates from observations?
- Figure 2 Labels need to be much larger, and a lot of white space can be removed from both subfigures. Could a 'zoom in' panel be provided near the hopf point to allow the detail there to be seen.
- Line 195 was other continuation software tested as well? Is the hopf super- or sub-critical? It is concerning to me that this cannot be identified/ is not a robust result.
- Figure 3 Is there not a 4$^{th}$ region between the hopf and saddle? However small, I think this should be acknowledged.
- Line 240 Why were no points with higher-than-reference salinity tested? Is there a physical justification?
- Line 400 The two sentences at the start of this section seem to contradict each other? Of course, with enough time and noise, any system would collapse/ recover. If the system is tipping because of noise fairly often (even for short times), I would say it is fairly unstable, and possibly near to a tipping threshold.
- Given the presence of a limit cycle, has the possibility of phase tipping been considered?

---

## Referee Comment (RC2)

**egusphere-2025-2074**

**1 Summary**

The manuscript *The effect of noise on the stability of convection in a conceptual model of the North Atlantic subpolar gyre* explores the sensitivity of the subpolar gyre (SPG) convection to noise by adapting and building on the conceptual model of Born and Stocker (2014, hereafter BS14). The BS14 model is non-dimensionalized and made autonomous in order to perform a bifurcation analysis. In addition, stochastic noise is added to represent variability in surface current's salinity and in freshwater forcing. The bifurcation analysis shows two stable states in the system: convective and non-convective, where the non-convective state is defined by the total non-dimensional volumetric transport in the gyre $M \leq 22$ Sv. The stability of the model is explored by analysing the sensitivity of convection to noise in salinity and freshwater forcing. It is found that the salinity noise impacts convection in the gyre significantly more than the noise in freshwater forcing. Additionally, it is found that the SPG recovers from the non-convective state across all tested parameters, a result which is not commonly found in Earth System Model (ESM) studies.

The quality of the scientific analysis in the manuscript is good and the topic is both important and thematically suitable for the Earth System Dynamics Journal. However, the study could go further in terms of the impact of the results. In the present form, the novelty of the research presented in the manuscript is questionable. This review presents some suggestions as to how the authors could push their study further.

**2 General comments**

The current description of the conceptual model is lacking. At multiple instances, the authors provide citations to previous work without outlining how these choices fit into the current model (for example, ll. 125-127: how is the value for $\tau_X$ picked?; ll. 445-448 what is the effect of picking $k \gg 1$ on the model?). BS14 provide an extensive discussion on the origin and physical meaning of the conceptual model parameters. Since this model is adapted in the current work, such in-depth discussion is not necessary - but sentence summary for different model parameters would greatly improve the transparency and clarity of the text. For example, mentioning that $U_{btp}$ corresponds to the volumetric transport of 20 Sv would be useful.

The choices made in connection with extending the BS14 model should also be clarified. On which basis were the values for $\tau_S$ and $\tau_F$ picked? What is the relation between parameters $c^*$ in BS14, and $c_1$ and $c_2$ in the adapted version of the model? An alternative mechanism for convective mixing is introduced without sufficient justification or description. How is the value for $c_2$ chosen?

The discussion about the realism of the model is somewhat contradictory throughout the text. In the model description, the amplitude of the noise is described as unrealistic and the choice is motivated by exploring the mechanistic aspects of the system (ll. 136-140). In the discussion, the noise values are instead described as "on the high end of realistic values" (ll. 355-356). I agree with the authors that the realistic frequency of the non-convective state under the current oceanographic conditions can be seen as an argument for the robustness of the model and the magnitude of the noise parameters used. The discussion on this aspect of the model could be streamlined throughout the text.

Section 4 does not convey that the results contribute significantly to the understanding

of the dynamics of the SPG. It is not obvious to me that the study goes far enough beyond the analysis of the conceptual model dynamics in the BS14 paper. One of the main results of the study is that the SPG convection is more sensitive to noise in the gyre salinity compared to freshwater forcing. However, as the authors themselves point out, this may be due to the structure of the conceptual model (ll. 339-340). Could the robustness of this result be tested in additional experiments? Another main result of the study is the resilience of the SPG convective state. The collapse and recovery of the SPG has been observed in at least one ESM study (Jochum et al. 2012). The physical mechanism which allows SPG to recover in the ESM is the freshwater flux through the Bafflin Bay. This and other ESM studies of the SPG dynamics could serve as a basis for a more exhaustive discussion on the physical meaning of the results in the present manuscript, and perhaps aid to design additional experiments that push the exploration of the idealized BS14 model with the inclusion of noise tipping further.

**3   Minor comments**

Larger figure labels would improve readability.

Punctuation should be edited throughout the text.

ll. 69-71: Why is it worrisome? Clarifying the magnitude of the SPG effect on the AMOC here would strengthen this statement.

l. 311: influence $\rightarrow$ influences

ll. 325-326: Is this not just due to the form of the equation of state ($\beta > \alpha$)?

Table A1: $r$ as a symbol for radius of the inner box and ratio of the surface and deep box heights should be distinguished; $S_4$ is the salinity of the deep gyre box.

**4   References**

Jochum, Markus, et al. "True to Milankovitch: Glacial inception in the new community climate system model." Journal of Climate 25.7 (2012): 2226-2239.

---

## Author Comment (AC1)

We thank the reviewer for their careful review and for providing thoughtful feedback. Below, we first respond (in blue) to the general comments, after which we provide detailed replies to the specific comments.

**General comments**

*This study uses a conceptual box model of the sub polar gyre to perform a bifurcation analysis and study noise-induced tipping. They find this model to be bi-stable, and test tipping from different initial conditions in phase space. As expected, the initial parametrisation and noise strength contribute to the likelihood of collapse in the time window observed. While an interesting model of the SPG, the model details and the implementation of noise needs to be much better explained and justified. I find that the results presented can be interpreted as simply confirming already known results of noise processes, and may not have such high impact. The grammar, equation, and some figure presentations need to be revised.*

**Reply:** We agree with the reviewer's comments about the description of the model and the implementation of the noise – in the current manuscript, this is indeed somewhat lacking and we will make the required changes to explain and motivate the model and noise better. We provide answers to the reviewer's specific comments about the noise and model parameters in the detailed comments below.

We also agree that many of our results can be expected from already known results of more general noise processes. However, we do believe that our results can be of use for the SPG modelling community. The collapse of convection in the SPG region is often presented as permanent (e.g. Armstrong McKay et al. (2022), Sgubin et al. (2017), Swingedouw et al. (2022)). This conclusion is based on modelling studies with Earth System Models (ESMs). We hope that indicating the possibility of non-permanent collapse in a conceptual model can provide useful context to these model studies. We will make this more clear in the revised manuscript.

**Specific comments**

*Section 2.1 I have difficulty understanding how the noise has been added to the model equations. First, it is not clear to me why there are not dynamical equations for boxes 2 and 4- this should be explained.*

**Reply:** In the Born and Stocker model (2014, hereafter: BS14), convection can only occur in the inner two boxes (1 and 3) of the model, that is, it assumes that convection occurs in the center of the gyre. The temperature and salinity in the gyre center are affected by A) the hydrographic properties of the surface current, B) the hydrographic properties of the deep current, and C) the atmosphere. A, B, C are treated as prescribed conditions for the variables in the center of the sea and are set constant at values of T, S, $T_0$ and F that are estimated from observations (BS14). Box 2 and 4 thus provide boundary conditions for boxes 1 and 3 and are therefore not represented by dynamical equations, much like the atmosphere is not represented by dynamical equations, but only by the terms $T_0$ and F.

**Changes in text:** We will clarify that the values of T, S, $U_s$ and $U_d$ in boxes 2 and 4 are boundary conditions and not dynamic variables in the model.

*Then, since there is no dynamic equation for box 2, the noise on $S_2$ is added to Equation 1d. I understand that since the noise is applied as an OU process, these noise processes have different time scales and will represent different physical processes. However, the way the noise is presented in the equations, this still appears to me as two noise processes added to box 1, not one on each box (1 and 2), therefore all noise processes would affect just box 1.*

**Reply:** The reviewer is correct in noting that both noise processes affect box 1. In our adjusted model, we apply noise to different variables. Ultimately all noise processes affect box 1, but due to the different nature of the parameters F and $S_2$ (see also our answer to the question below), we apply the noise to these two variables separately.

**Changes in text:** We will clarify that the noise is added to the parameters $S_2$ and F and not to the boxes as such.

*The amplitude of the noise term with zeta_S will also be heavily influenced by the pre-factors, and it is not clear to me that this is fully taken into account.*

**Reply:** This is correct. The prefactors are a result of the relative importance of the physical mechanism between the box-2 salinity and freshwater terms in the salinity budget of box 1. Varying F directly increases or decreases the amount of freshwater that's added to box 1. By comparison, varying $S_2$ only indirectly changes the salinity of box 1 by first changing the strength of the baroclinic current $U_s$, and then the magnitude of horizontal eddy transport $mu_H*U_s*(S_2-S_1)$. This is a highly nonlinear process which reflects the physical mechanism by which salinity anomalies are transported from the boundary current to the convective core of the gyre. To some extent, we take this difference into account by choosing different values for the noise amplitude sigmaS and sigmaF (Table 2).

As the noise is added to the parameters $S_2$ and F, all pre-factors that apply to $S_2$ therefore also apply to the noise term zeta_S. We discuss this to some extent in in Sect. 4.2 (l. 336 – 346).

**Changes in text:** In l. 135 add "We note that $S_2$ and consequently the noise term zeta_S are influenced by nonlinear prefactors in a way that F and zeta_F are not. This is discussed in Sect. 4.2."

*Similarly, why would precipitation only affect the surface gyre current and not the surface core box? What does 'precipitation upstream' mean (l.175)?*

**Reply:** Precipitation affects both the surface gyre current and the surface core box. In fact, the freshwater flux F mostly represents precipitation (l. 89). The salinity of the surface gyre current $S_2$ is influenced by the salinity of the currents in the Nordic and Irminger Seas and the Baffin Bay, sea ice melt and export from these regions, and Greenland Ice Sheet meltwater. 'Precipitation upstream' refers to precipitation that affects the salinity of the aforementioned currents, but we agree that this term is unclear.

**Changes in text**: We will revise the text throughout to make clear what physical processes contribute to (variability in) the values of F and $S_2$. We will replace the term 'precipitation upstream' with 'precipitation'.

*Section 3.3 It is not clear to me that the collapses discussed are full tipping events? Are these full transitions to the alternative state? In the discussion lines ~250-270. It is noted 'even in the least stable case... the gyre never becomes fully nonconvective'. It is therefore not clear to me that a full tipping event has taken place? What does the alternative stable state correspond to, and therefore is this an expected result? Given the bifurcation diagram is found, could one not check if the system has actually transitioned to the alternative state and is in the alternative basin of attraction?*

**Reply:** We thank the reviewer for questioning this. To confirm that the model is indeed transitioning to the alternative state, we ran additional simulations. We ran time integrations with noise in the gyre salinity S_2 (identical to those described in the current manuscript) and identified events when the gyre transport M was below a threshold of 21 Sv (see discussion below) for multiple consecutive years.

We then branched off from this simulation twice for each event. First, we ran a deterministic simulation with no noise using the values of T1, S1, T3, S3 from the last year of the event as initial conditions. We also ran a deterministic simulation using the values of T1, S1, T3, S3 from one year after the first branched-off simulation as initial conditions. These simulations are shown below for four events (for parameter set B):

[Figure]

Clearly, when the gyre transport is below the threshold value, it is in the alternate basin of attraction and therefore moves to the constant non-convective (low-transport) state. We conclude from this analysis that the collapses we discuss are full transitions between states.

We note that defining the threshold is not easy. In our manuscript, we used a threshold value of 22 Sv. However, when running the branched deterministic simulations, we found that this value is too high; when $21 < M < 22$, sometimes the gyre is still in the basin of attraction of the convective state and therefore returns to high values of M. This can be seen in the bifurcation diagram (Fig. 2a). Finding the exact value of the threshold is, unfortunately, computationally expensive. We propose erring on the side of caution and taking a threshold value of 21 Sv. This means our statistics will underestimate the amount of years in which the gyre is in a non-convective state (see panel d in the figure above), but we expect this difference to be minor, since only one or two years without convection will not be counted in only a few cases.

A last note is that for the deterministic simulations for parameter set A (see Fig. 3) the gyre always returns to the convective high-transport state. This is expected since for these parameters the gyre is in the mono-stable regime, and the non-convective regime is unstable. Noise on S2 can temporarily allow for a transition to the bi-stable regime, but as soon as it is removed the system returns to the convective state.

We thank the reviewer for this feedback, since it has improved our analysis.

**Changes in text:** We will run the analysis in Sect. 3.3 again with a threshold of 21 Sv instead of 22 Sv and change the figures, text and tables accordingly. We will clarify that branched-off simulations show that we observe full state transitions for parameter sets B-F, but not for A and interpret the variability for parameter set A as noise events followed by a recovery.

*The results discussed on page 11, I think are expected mathematically. As you move the initial conditions in phase space, you essentially start the simulations with different effective potential barrier heights for transitions. Therefore, with the same noise amplitude and a fixed time, a different percentage of transitions will take place according to large deviation theory (Freidlin & Wentzell, 1984, Bouchet & Reygner, 2016).*

**Reply:** We agree with the reviewer that these results are expected mathematically. However, we do believe that presenting these results in the context of the SPG can be insightful for the SPG modelling community.

**Changes in text:** We will frame these results in the context of large deviation theory as described by Freidlin & Wentzell (1984) and Bouchet & Reygner (2016). We will also add a reference to Kuhlbrodt & Monahan (2003) who discuss this in the context of convection.

*Additionally, in some of these 'short excursions' if there is not a full transition to the alternative state, this would then be an example of a noise event followed by a noise-induced recovery (Chapman, Ashwin et al 2024). This could be checked, and the tipping (or not) mechanisms could be identified since the bifurcation diagram is known.*

**Reply:** See discussion above. We will discuss the noise event followed by noise-induced recovery we observe for parameter set A in the context of Chapman, Ashwin et al. (2024).

*Grammar and phrasing needs revising throughout.*

**Reply:** We will critically revise the grammar and phrasing throughout the manuscript.

*Model equations need to be defined more rigorously, not all variables are defined, should be defined immediately after the Equations.*

**Reply:** We will define all variables immediately after the Equations.

*Where do the values of the model parameters come from? Literature, GCMs, physical estimates from observations?*

**Reply:** All model parameters are taken from BS14, who estimated the values based on observations. We will clarify this in the text.

**Changes in text:** In the description of Table 1 change "The values were calculated from the default model parameters as outlined in Table 1 of Born and Stocker (2014) and Appendix A. No values are given for the parameters mu_h, mu_C, and mu_A, as these values do not have an intuitive interpretation."

to

Suggestion: "We use the parameter values in Born and Stocker (2014) (their Table 1, here Table A1), to compute the parameter values for our non-dimensionalised model. Their parameter estimates are based on observations and expert assessment. The non-dimensionalisation introduces some additional dimensionless parameters (eta, mu_h, mu_C and mu_A), for which no dimensional values are given for lack of interpretability."

*Figure 2 Labels need to be much larger, and a lot of white space can be removed from both subfigures. Could a 'zoom in' panel be provided near the hopf point to allow the detail there to be seen.*

**Reply:** We thank the reviewer for this suggestion. We will increase the font size in figures throughout the manuscript and remove white space where possible. We will add a "zoom in" figure (see below) in the appendix for interested readers, as we think that adding such a figure in the main text will distract from the main message.

[Figure]

*Line 195 was other continuation software tested as well? Is the hopf super- or subcritical? It is concerning to me that this cannot be identified/ is not a robust result.*

**Reply:** We found the Hopf bifurcation in the experiments with S2 to be control parameter as supercritical and the Hopf bifurcation in the experiment with F as control parameter to be subcritical. Although we searched for periodic orbits for both points, we did not find them, possibly because their amplitudes are very small. It is not possible to dismiss this bifurcation point as a numerical error. It is found in almost all continuations with S2 and F, and in codim-2 continuations (manuscript Fig. 3) the presence of a Hopf point close to a saddle node is extremely constant. Since we did not find any periodic orbits in our time integrations, these Hopf bifurcations do not affect the results. For this reason, no other continuation software was tested.

It's worth nothing that bifurcation analysis of conceptual models describing the AMOC often shows the existence of Hopf bifurcations that are close to saddle node bifurcations. Titz et al. (2002b) found that when freshwater flux is increased in the four-box interhemispheric model described by Rahmstorf (1996), the upper stable branch loses its stability in a Hopf bifurcation, which is followed by a saddle node linking two unstable branches. It can be shown that this Hopf bifurcation is always subcritical and that as such, all periodic orbits emerging from this point are unstable (Titz et al., 2002a). These results were replicated in the five-box model described in Wood et al. (2019) by Alkhayuon et al. (2019), who also found that the distance between the Hopf and saddle node bifurcations increases with increasing atmospheric $CO_2$ concentrations. Furthermore, the presence of a subcritical Hopf point near a saddle node is also found in hosing experiments with ESMs (van Westen et al., 2024), indicating that this result is not unique to models of low and intermediate complexity.

Since the Hopf bifurcations do not affect our results, we will keep the discussion of these points to a minimum in the manuscript, but elaborate on the above mentioned points in the SI for the interested readers.

**Changes in text:** We will add the zoomed figure and a discussion on the Hopf bifurcations to the appendix and in ll. 193-196 we change

"In our analysis, we found a Hopf bifurcation very close to a saddle node bifurcation for a wide range of parameters. However, attempts at finding the corresponding periodic orbits were unsuccessful and it is not clear if these Hopf bifurcations are real 195 characteristics of the model or rather artefacts of the used continuation software. Consequently, these points are shown in the results, but will not be discussed further."

to

Suggestion: "In our analysis, we found a Hopf bifurcation very close to a saddle node bifurcation for a wide range of parameters. This feature together with zoomed-in figures

of the region is discussed in the SI. Since these Hopf bifurcations do not affect the results, they will not be discussed further."

*Figure 3 Is there not a 4th region between the hopf and saddle? However small, I think this should be acknowledged.*

**Reply:** The reviewer is correct in noting that this region exists. However, we would argue that the effect of this small region is negligible, also since no periodic orbits have been found. Nonetheless, we agree it is good to acknowledge its existence and will add a note on its existence, without "naming" it like the others.

**Changes in text:** In l. 225 we will add "Formally, a fourth region can be distinguished between regions II and III. This region is demarcated by the Hopf bifurcation and the lower saddle node bifurcation. However, this region is very small and since we do not find period orbits associated with the Hopf bifurcation, we do not consider it in our analysis."

*Line 240 Why were no points with higher-than-reference salinity tested? Is there a physical justification?*

**Reply:** No points with higher-than-reference salinity were tested, because nearly all external changes to the gyre current (increase in meltwater from the Greenland Ice Sheet, increases in precipitation, increase in Fram Strait sea ice export) serve to decrease its salinity. In addition, observations also indicate that the North Atlantic is freshening (de Steur et al., 2018).

Moreover, points with higher-than-reference salinity mostly fall into the monostable regime (Fig. 3), so we don't expect an analysis of such points to add much to the results.

**Changes in text:** We will add a short discussion on mechanisms that can decrease the salinity of the gyre region in the introduction. In l. 242 add

"No points with higher-than-reference salinity were tested, since nearly all external changes to the gyre current serve to decrease its salinity."

*Line 400 The two sentences at the start of this section seem to contradict each other? Of course, with enough time and noise, any system would collapse/ recover. If the system is tipping because of noise fairly often (even for short times), I would say it is fairly unstable, and possibly near to a tipping threshold.*

**Reply:** We thank the reviewer for pointing this out, the phrasing we used here is indeed unclear. The point we want to make is that that the system can change between a state with and without convection quite easily, and that a collapse of convection does not have to be permanent. This is in contrast with the common result that a collapse of convection in the SPG is a more or less irreversible change with grave consequences. This conclusion is of course based on several assumptions. Most notably, in deriving these results we have assumed that the atmospheric temperature T0 remains constant. It is possible that the stability of convection can change in this model as other climate parameters (e.g. T0, Ubtp) vary. This is an interesting avenue for future research.

**Changes in text:** In l. 400 change

"Based on the results presented here, it can be concluded that convection in the North Atlantic subpolar gyre is quite stable under current oceanographic conditions in a simple model"

to

"Based on the results from the simple model presented here, we conclude that a permanent collapse of convection in the North Atlantic subpolar gyre is unlikely under current oceanographic and atmospheric conditions."

*Given the presence of a limit cycle, has the possibility of phase tipping been considered?*

**Reply:** Indeed, mathematically the limit cycle should exist, but as we noted above, we did not find any periodic orbits, possibly because the amplitude is very small. Hopf bifurcations close to saddle node bifurcations are frequently found in such box models of the ocean, and clearly have something to do with the destabilization process. However, it is difficult to attach physical mechanisms to this behavior in the current context. Therefore we prefer to not overinterpret the mathematical findings in this model.

**References**

Alkhayuon, H., Ashwin, P., Jackson, L. C., Quinn, C., & Wood, R. A. (2019). Basin bifurcations, oscillatory instability and rate-induced thresholds for Atlantic meridional overturning circulation in a global oceanic box model. *Proceedings of the Royal Society A: Mathematical, Physical and Engineering Sciences*, *475*(2225), 20190051. https://doi.org/10.1098/rspa.2019.0051

Armstrong McKay, D. I., Staal, A., Abrams, J. F., Winkelmann, R., Sakschewski, B., Loriani, S., Fetzer, I., Cornell, S. E., Rockström, J., & Lenton, T. M. (2022). Exceeding 1.5°C global warming could trigger multiple climate tipping points. *Science*, *377*(6611), eabn7950. https://doi.org/10.1126/science.abn7950

Born, A., & Stocker, T. F. (2014). Two Stable Equilibria of the Atlantic Subpolar Gyre. *Journal of Physical Oceanography*, *44*(1), 246–264. https://doi.org/10.1175/JPO-D-13-073.1

de Steur, L., Peralta-Ferriz, C., & Pavlova, O. (2018). Freshwater Export in the East Greenland Current Freshens the North Atlantic. *Geophysical Research Letters*, *45*(24), 13,359-13,366. https://doi.org/10.1029/2018GL080207

uhlbrodt, T., & Monahan, A. H. (2003). Stochastic Stability of Open-Ocean Deep Convection. *Journal of Physical Oceanography*, *33*(12), 2764–2780. https://doi.org/10.1175/1520-0485(2003)033<2764:SSOODC>2.0.CO;2

Li, H., & Fedorov, A. V. (2021). Persistent freshening of the Arctic Ocean and changes in the North Atlantic salinity caused by Arctic sea ice decline. *Climate Dynamics*, *57*(11), 2995–3013. https://doi.org/10.1007/s00382-021-05850-5

Rahmstorf, S. (1996). On the freshwater forcing and transport of the Atlantic thermohaline circulation: *Climate Dynamics*, *12*(12), 799–811. https://doi.org/10.1007/s003820050144

Sgubin, G., Swingedouw, D., Drijfhout, S., Mary, Y., & Bennabi, A. (2017). Abrupt cooling over the North Atlantic in modern climate models. *Nature Communications*, *8*(1), 14375. https://doi.org/10.1038/ncomms14375

Swingedouw, D., Bily, A., Esquerdo, C., Borchert, L. F., Sgubin, G., Mignot, J., & Menary, M. (2021). On the risk of abrupt changes in the North Atlantic subpolar gyre in CMIP6 models. *Annals of the New York Academy of Sciences*, *1504*(1), 187–201. https://doi.org/10.1111/nyas.14659

Titz, S., Kuhlbrodt, T., & Feudel, U. (2002a). Homoclinic bifurcation in an ocean circulation box model. *International Journal of Bifurcation and Chaos*, *12*(04), 869–875. https://doi.org/10.1142/S0218127402004759

Titz, S., Kuhlbrodt, T., Rahmstorf, S., & Feudel, U. (2002b). On freshwater-dependent bifurcations in box models of the interhemispheric thermohaline circulation. *Tellus A, 54*(1), 89–98. https://doi.org/10.1034/j.1600-0870.2002.00303.x

Van Westen, R. M., Jacques-Dumas, V., Boot, A. A., & Dijkstra, H. A. (2024). The Role of Sea Ice Insulation Effects on the Probability of AMOC Transitions. *Journal of Climate, 37*(23), 6269–6284. https://doi.org/10.1175/JCLI-D-24-0060.1

---

## Author Comment (AC2)

We thank the reviewer for their careful review and for providing useful suggestions on how to improve this manuscript. Below, we first respond to the general comments, after which we provide detailed replies (in blue) to the specific comments.

*The manuscript The effect of noise on the stability of convection in a conceptual model of the North Atlantic subpolar gyre explores the sensitivity of the subpolar gyre (SPG) convection to noise by adapting and building on the conceptual model of Born and Stocker (2014, hereafter BS14). The BS14 model is non-dimensionalized and made autonomous in order to perform a bifurcation analysis. In addition, stochastic noise is added to represent variability in surface current's salinity and in freshwater forcing. The bifurcation analysis shows two stable states in the system: convective and non-convective, where the non-convective state is defined by the total non-dimensional volumetric transport in the gyre M ≤ 22 Sv. The stability of the model is explored by analysing the sensitivity of convection to noise in salinity and freshwater forcing. It is found that the salinity noise impacts convection in the gyre significantly more than the noise in freshwater forcing. Additionally, it is found that the SPG recovers from the non-convective state across all tested parameters, a result which is not commonly found in Earth System Model (ESM) studies. The quality of the scientific analysis in the manuscript is good and the topic is both important and thematically suitable for the Earth System Dynamics Journal. However, the study could go further in terms of the impact of the results. In the present form, the novelty of the research presented in the manuscript is questionable. This review presents some suggestions as to how the authors could push their study further.*

**General comments**

*The current description of the conceptual model is lacking. At multiple instances, the authors provide citations to previous work without outlining how these choices fit into the current model (for example, ll. 125-127: how is the value for $\tau X$ picked?; ll. 445-448 what is the effect of picking $k \gg 1$ on the model?).*

**Reply:** We thank the reviewer for pointing this out. In an earlier version of the manuscript the reference to Dijkstra et al. (2023) (l. 126) was made to justify the use of an Ornstein-Uhlenbeck process. Later, we added the more relevant references to Penland and Ewald, 2008; Boers et al., 2022; Ditlevsen and Johnsen, 2010 in l. 122-123. These more relevant references make the reference to Dijkstra et al. (2023) redundant and we will therefore remove this reference from the manuscript. The choice of tau_X in the Ornstein-Uhlenbeck processes is motivated in the next paragraph in the manuscript (ll. 129 – 135, see also the discussion below).

Picking k >> 1 ensures that boxes 1 and 3 immediately mix if the stratification is unstable, in line with the original BS14 model. A lower value of k would result in slower mixing. Picking k >> 1 is a common choice to implement a step function analytically (Dijkstra, 2004, pp. 70).

**Changes in text**: we will remove the reference to Dijkstra (2023) and in l. 447 change "where $k \gg 1$ (e.g. Dijkstra, 2004). In this study a value of $k = 10^5$ was used"

to

"where k >> 1 to ensure mixing between the boxes occurs instantaneously (e.g. Dijkstra, 2004, pp. 70). In this study a value of $k = 10^5$ was used."

We will also define all variables immediately after the Equations and expand our discussion of the parameters (see below).

*BS14 provide an extensive discussion on the origin and physical meaning of the conceptual model parameters. Since this model is adapted in the current work, such in-depth discussion is not necessary – but sentence summary for different model parameters would greatly improve the transparency and clarity of the text. For example, mentioning that Ubtp corresponds to the volumetric transport of 20 Sv would be useful.*

**Reply:** We agree that the current presentation of the model parameters in the text is lacking and will extend it.

**Change in text:** In l. 120 we change

"All other parameters are prescribed."

to

"All other parameters are prescribed: $U_{btp}$ is the barotropic component of the current and has a strength of 20 Sv, r represents the ratio of the surface and deep box height, eta the strength of the thermal wind, $mu_H$ the horizontal mixing efficiency, $mu_C$ the convection efficiency, and $mu_A$ the atmosphere-ocean exchange efficiency."

In the description of Table 1 we change

"The values were calculated from the default model parameters as outlined in Table 1 of Born and Stocker (2014) and Appendix A. No values are given for the parameters eta, $mu_H$, $mu_C$, and $mu_A$, as these values do not have an intuitive interpretation."

to

Suggestion: "We use the parameter values in Born and Stocker (2014) (their Table 1, here Table A1), to compute the parameter values for our non-dimensionalised model. Their parameter estimates are based on observations and expert assessment. The non-dimensionalisation introduces some additional dimensionless parameters (eta, $mu_H$, $mu_C$, and $mu_A$), for which no dimensional values are given for lack of interpretability."

*The choices made in connection with extending the BS14 model should also be clarified. On which basis were the values for τS and τF picked?*

**Reply:** We selected the values of tau_S and tau_F to represent the different time scales of variability of ocean and atmosphere (ll. 132 – 135). Changes in the gyre salinity propagate on a time scale of several years (e.g. the Great Salinity Anomalies), whereas changes in the freshwater flux happen on a shorter (seasonal) timescale. For clarity we will rephrase the corresponding lines in the manuscript.

**Change in text:** In l. 132-135 we change

"To simulate the different intrinsic time scales of variability in ocean and atmosphere, correlation time scales of tau_S =1 yr and tau_F = 90 days were used unless specified otherwise. With these time scales, the stochastic variations in gyre salinity S2 (described by zeta_S) can be interpreted as being driven by external variations in for example sea ice cover, and the stochastic variations in freshwater forcing F (described by zeta_F) as quasi-seasonal variations in precipitation."

to

"We select the correlation timescales tau_S and tau_F such that the noise processes represent the time scales of variability in ocean and atmosphere; tau_S =1 yr and tau_F = 90 days (unless specified otherwise). This means that changes in the gyre salinity have a timescale of years, corresponding to that of the Great Salinity anomalies and driven by e.g. sea ice cover variations. Changes in the freshwater forcing correspond to quasi-seasonal precipitation variability."

*What is the relation between parameters c ∗ in BS14, and c1 and c2 in the adapted version of the model? An alternative mechanism for convective mixing is introduced without sufficient justification or description. How is the value for c2 chosen?*

**Reply:** Parameter c in BS14 corresponds to c1 in our model, with c1 = c* A / V (BS14 eq. 15) and c* = 0.03 denotes the mixing efficiency.

We parameterize convection analytically, rather than handling it computationally at every time step (i.e. mix when the box 1 density exceeds that of box 3). Conceptually, the analytic convection term contains the terms.

 [strength of convection] * [step function that ensures convection only occurs under unstable stratification] * [difference in T/S].

Because we take a high value of k (see discussion above), this approach is equivalent to that of BS14. The difference is that our model does not contain a conditional step in the integration and therefore can be studied with continuation software.

Term c2 in our equations represents the strength of convection. We determine the value of c2 in relation to that of c1, since it is the relative importance of the horizontal and convective mixing terms that matters. We choose c2 such that the ratio c2/c1 = 10^3 to ensure box 1 and 3 are mixed very fast, staying close to the instantaneous mixing of BS14. This choice does not affect the bifurcation structure of the model, but only the time scale at which the steady state solution is reached.

Summarizing, instead of a discrete (conditional) mechanism for convective mixing as in BS14, we consider a continuous one by using a step function. We take the values of k and c2 such that this continuous mechanism is as close to the original BS14 model as possible.

**Changes in text**: In l. 441 we add

"Note that the (dimensional) parameter c_1 in our formulation is equal to c in BS14, such that c_1 =  c* A / V."

In l. 452 we add

"Taking high values for both c_2 and k ensures that convective mixing between box 1 and 3 happens nearly instantaneously, in line with the implementation of convection in BS14."

We also add the values of c*, A, V to Table A1.

*The discussion about the realism of the model is somewhat contradictory throughout the text. In the model description, the amplitude of the noise is described as unrealistic and the choice is motivated by exploring the mechanistic aspects of the system (ll. 136-140). In the discussion, the noise values are instead described as "on the high end of realistic values" (ll. 355-356). I agree with the authors that the realistic frequency of the non-convective state under the current oceanographic conditions can be seen as an argument for the robustness of the model and the magnitude of the noise parameters used. The discussion on this aspect of the model could be streamlined throughout the text.*

**Reply:** We thank the reviewer for pointing this out. We will streamline the discussion and use "on the high end of realistic values" to describe the noise consistently throughout.

*Section 4 does not convey that the results contribute significantly to the understanding of the dynamics of the SPG. It is not obvious to me that the study goes far enough beyond the analysis of the conceptual model dynamics in the BS14 paper. One of the main results of the study is that the SPG convection is more sensitive to noise in the gyre salinity compared to freshwater forcing. However, as the authors themselves point out, this may be due to the structure of the conceptual model (ll. 339-340). Could the robustness of this result be tested in additional experiments?*

**Reply:** The structure of the conceptual model is such that it best represents the physical mechanisms relevant for the gyre variability. Hence, changing the structure of the model would mean it no longer accurately represents the mechanisms of freshwater forcing and gyre salinity changes. It is encouraging to see that with such a simple representation of the mechanisms results that are in line with observations can be found, i.e. that convection is more sensitive to variability in the gyre salinity than precipitation.

There is ample evidence that supports the hypothesis that convection in the Labrador sea is more sensitive to noise in the gyre salinity (brought in via the boundary currents) than noise in the freshwater forcing. For example, Yashayaev (2024) identified the freshening of the Labrador sea, caused by a release of low-salinity water from the Beaufort Gyre, as the main cause of the 2023 convective shutdown. Similarly, Gelderloos et al. (2012) argued that the Great Salinity Anomaly (a blob of anomalously low salinity moving through the region) initiated the shutdown of convection in 1969. In addition, they found the advection of saltier waters to be one of the reasons convection restarted in 1972. Conversely, we are not aware of analyses that show noise in the atmospheric freshwater forcing (i.e., precipitation in the center of the Labrador sea) contributing meaningfully to the variability of convection here.

It is worth noting here that the different response to noise in the gyre salinity and the freshwater forcing is not unique to the noise, but is a result of the difference in physical mechanism between the freshwater flux and gyre salinity terms in the salinity budget of $S_1$. Varying F directly increases or decreases the amount of freshwater that's added to box 1. By comparison, varying $S_2$ only indirectly changes the salinity of box 1 by first changing the strength of the baroclinic current $U_s$, and then the magnitude of horizontal eddy transport $\mu_H * U_s * (S_2 - S_1)$. This is a highly nonlinear process which reflects the physical mechanism by which salinity anomalies are transported from the boundary current to the convective core of the gyre.

**Changes in text**: We will add a discussion of observed shutdowns of deep convection in the Labrador sea, how these shutdowns often depend on changes in the salinity of the boundary current (and not on changes in freshwater forcing) and how our conceptual model adds to the mechanistic understanding of these observed shutdowns.

*Another main result of the study is the resilience of the SPG convective state. The collapse and recovery of the SPG has been observed in at least one ESM study (Jochum et al. 2012). The physical mechanism which allows SPG to recover in the ESM is the freshwater flux through the Bafflin Bay. This and other ESM studies of the SPG dynamics could serve as a basis for a more exhaustive discussion on the physical meaning of the results in the present manuscript, and perhaps aid to design additional experiments that push the exploration of the idealized BS14 model with the inclusion of noise tipping further.*

**Reply:** We thank the reviewer for this suggestion and will incorporate this reference. Deep convection in the Labrador Sea is also intermittent in a 12-model historical ensemble of EC-Earth (Brodeau and Koenigk, 2016), showing that at least this ESM is in principle able to simulate a recovery of convection in the SPG region. In addition, convection in the region has been observed to stop and restart in at least one CMIP6 model (NorESM-LM) (Swingedouw et al, 2021).

Unfortunately, the results from most ESM studies on the SPG region and collapse of cannot be compared directly to our results. We assume a constant background climatic state (i.e. $T_0$ and $U_{btp}$ are constant), whereas most relevant ESM modelling studies study the response of the SPG region to a changing background state. Assuming this constant background state we conclude that convection in the SPG is stable to salinity perturbations in our current climate. It is possible that the stability of convection can change in this model as other climate parameters (e.g. $T_0$, $U_{btp}$) vary. This is an interesting avenue for future research.

**Changes to text:** We will add a discussion of ESM studies that show the intermittency of convection in the SPG to show that this result is not unique to the simple model we use, while stressing that direct comparison between this work and most ESM modelling work is not possible.

**3 Minor comments**

*Larger figure labels would improve readability.*

**Reply:** We thank the reviewer for pointing this out and will increase the label size for all figures.

*Punctuation should be edited throughout the text.*

**Reply**: We will critically revise the punctuation, grammar and phrasing throughout the manuscript.

*ll. 69-71: Why is it worrisome? Clarifying the magnitude of the SPG effect on the AMOC here would strengthen this statement.*

**Reply:** We thank the reviewer for this suggestion and will clarify the link between the SPG and the AMOC.

**Changes to text**: In l.71 we add

"In addition, there are indications that a collapse of convection in the SPG can precede an AMOC collapse (Danabosoglu et al. (2016), Drijfhout et al. (2025))."

*l. 311: influence → influences*

**Reply:** We will correct this.

*ll. 325-326: Is this not just due to the form of the equation of state ($\beta > \alpha$)?*

**Reply**: This is an interesting point, although it is difficult to compare alpha and beta directly since their units are different and they are multiplied with variables (S and T) of different magnitude. The equation of state we used here is often used in idealized models. If the equation of state indeed causes strong dependence on noise in the salinity in our model, other studies using box models of the AMOC with noise (which often use the same equation of state) should show a similar dependence. We are not aware of similar results in such studies and have found no reason to believe the equation of state is causing the stronger sensitivity to haline perturbations.

*Table A1: r as a symbol for radius of the inner box and ratio of the surface and deep box heights should be distinguished; S4 is the salinity of the deep gyre box.*

**Reply:** We thank the reviewer for pointing this out. We will define the radius of the inner box as r_sd instead of r and correct the definition of S4.

**References**

Brodeau, L., & Koenigk, T. (2016). Extinction of the northern oceanic deep convection in an ensemble of climate model simulations of the 20th and 21st centuries. *Climate Dynamics*, *46*(9), 2863–2882. https://doi.org/10.1007/s00382-015-2736-5

Danabasoglu, G., Yeager, S. G., Kim, W. M., Behrens, E., Bentsen, M., Bi, D., Biastoch, A., Bleck, R., Böning, C., Bozec, A., Canuto, V. M., Cassou, C., Chassignet, E., Coward, A. C., Danilov, S., Diansky, N., Drange, H., Farneti, R., Fernandez, E., … Yashayaev, I. (2016). North Atlantic simulations in Coordinated Ocean-ice Reference Experiments phase II (CORE-II). Part II: Inter-annual to decadal variability. Ocean Modelling, 97, 65–90. https://doi.org/10.1016/j.ocemod.2015.11.007

Drijfhout, S., Angevaare, J. R., Mecking, J., van Westen, R. M., & Rahmstorf, S. (2025). Shutdown of northern Atlantic overturning after 2100 following deep mixing collapse in CMIP6 projections. *Environmental Research Letters*, *20*(9), 094062. https://doi.org/10.1088/1748-9326/adfa3b

Gelderloos, R., Straneo, F., & Katsman, C. A. (2012). Mechanisms behind the Temporary Shutdown of Deep Convection in the Labrador Sea: Lessons from the Great Salinity Anomaly Years 1968–71. *Journal of Climate*, *25*(19), 6743–6755. https://doi.org/10.1175/JCLI-D-11-00549.1

Swingedouw, D., Bily, A., Esquerdo, C., Borchert, L. F., Sgubin, G., Mignot, J., & Menary, M. (2021). On the risk of abrupt changes in the North Atlantic subpolar gyre in CMIP6 models. *Annals of the New York Academy of Sciences*, *1504*(1), 187–201. https://doi.org/10.1111/nyas.14659

Yashayaev, I. (2024). Intensification and shutdown of deep convection in the Labrador Sea were caused by changes in atmospheric and freshwater dynamics. *Communications Earth & Environment*, *5*(1), 156. https://doi.org/10.1038/s43247-024-01296-9